# Integrated molecular characterisation of the MAPK pathways in human cancers reveals pharmacologically vulnerable mutations and gene dependencies

Musalula Sinkala [1✉], Panji Nkhoma [2], Nicola Mulder [1] & Darren Patrick Martin[1]

The mitogen-activated protein kinase (MAPK) pathways are crucial regulators of the cellular processes that fuel the malignant transformation of normal cells. The molecular aberrations which lead to cancer involve mutations in, and transcription variations of, various MAPK pathway genes. Here, we examine the genome sequences of 40,848 patient-derived tumours representing 101 distinct human cancers to identify cancer-associated mutations in MAPK signalling pathway genes. We show that patients with tumours that have mutations within genes of the ERK-1/2 pathway, the p38 pathways, or multiple MAPK pathway modules, tend to have worse disease outcomes than patients with tumours that have no mutations within the MAPK pathways genes. Furthermore, by integrating information extracted from various large-scale molecular datasets, we expose the relationship between the fitness of cancer cells after CRISPR mediated gene knockout of MAPK pathway genes, and their dose-responses to MAPK pathway inhibitors. Besides providing new insights into MAPK pathways, we unearth vulnerabilities in specific pathway genes that are reflected in the re sponses of cancer cells to MAPK targeting drugs: a revelation with great potential for guiding the development of innovative therapies.

[1] University of Cape Town, Cape Town, South Africa. [2] University of Zambia, Lusaka, Zambia. ✉email: smsinks@icloud.com

The mitogen-activated protein kinase (MAPK) pathways are crucial cell signal transduction pathways that regulate molecular processes such as cell proliferation, cell differentiation, cell survival, cancer dissemination, and resistance to drug therapy[1,2]. The MAPK pathways involve four main modules: the extracellular-signal-regulated kinase 1 and 2 (ERK1/2) pathway (also known as the classical pathway), the c-Jun N-terminal kinase (JNK) pathway, the p38 pathway and the ERK5 pathway[1,3]. Each of these modules is initiated by specific extracellular signals that lead to the sequential activation of a MAP kinase kinase kinase (MAPKKK), then a MAP kinase kinase (MAPKK) which phosphorylates a MAP kinase (MAPK)[1–3]. Subsequent phosphorylation of the MAP kinases results in the activation of multiple substrates, including the transcription factors that are effectors of cellular responses to MAPK pathway activation[1–4].

Over the last few decades, our understanding of the individual MAPK signalling modules and their role in oncogenesis has grown significantly and, along with this increased interest, concerted efforts to treat various tumours that display altered MAPK signalling[5–9]. We now have many anti-cancer drugs that target the components of the MAPK pathways, most of which have been successful in treating cancers of, among other tissues, the skin and kidney[10–13].

Several genetic aberrations, including mutations in, and copy number variations of, MAPK pathway genes have been identified in human cancers, and several of the proteins encoded by these genes are promising drug targets[14–17]. However, we still do not have a complete understanding of the extent to which MAPK pathways are altered across the entire spectrum of human cancers, and whether these alterations impact either disease outcomes or the responses of tumours to anti-MAPK pathway drugs.

There is, therefore, a pressing need to identify cancer types that harbour mutations in genes that encode MAPK pathway proteins. By extension, a better understanding of how genetic alterations and gene dependencies within cancer cells impact the responses of these cells to anti-cancer drugs will aid in rationally selecting the best available drugs for treating particular cancers. Furthermore, a better appreciation of the specific MAPK pathway aberrations in different human cancers would likely translate to improved disease outcome predictions because some of the genetic alterations of cancer cells are likely to be directly associated with disease aggressiveness and clinical outcomes.

The large-scale molecular profiling of human cancers such as has been carried out by The Cancer Genome Atlas (TCGA[18]) project and other consortia and compiled by the cBioPortal project[19], has yielded vast amounts of publicly accessible data that can be leveraged to understand the complexity of MAPK signalling in human cancers. Further, the Achilles project is processing and releasing data on gene dependencies in hundreds of cancer cell lines, as determined using high-resolution fitness screens of cells following the knockout of particular genes using CRISPR (Clustered Regularly Interspaced Short Palindromic Repeats)[20]. In addition, the Genomics of Drug Sensitivity in Cancer (GDSC[21]) project and the Cancer Cell Line Encyclopaedia (CCLE[22,23]) project has screened, and continue to screen, the sensitivity of thousands of human cancer cell lines to hundreds of diverse small molecule inhibitors.

The efforts of the TCGA, Achilles, GDSC and CCLE projects are complemented by the library of integrated cellular-based signatures (LINCS) project which aspires to illuminate the responses of complex cellular systems[24,25] to drug perturbation. The LINCS project provides datasets detailing the molecular and functional changes that occur in thousands of different human cell types following their exposure to thousands of drugs and/or genetic perturbations[25,26].

The large scales and the public accessibility of the datasets that these projects are producing provide new prospects for us to link cancer phenotypes to molecular features, clinical outcomes, and the drug responses of tumours. Here, by integrating information extracted from these datasets, we provide a comprehensive analysis of MAPK pathways across different cancer types. Besides providing novel biological insights into the mutational landscape of MAPK pathway genes, and how these affect disease outcomes in cancer patients, we show both the specific MAPK pathway genes that impact the fitness of cancer cells and how vulnerabilities exposed by mutations and transcriptional changes in these genes are reflected in the responses of cancer cells to MAPK pathway inhibitors.

## Results

**The mutational landscape of MAPK pathway genes**. Based on the available literature and the KEGG pathway database[27], we first identified a list of all genes encoding members of the ERK1/2, JNK, p38 and ERK5 pathways. This list consisted of 142 genes that function mainly through the MAPK pathways as core genes of the ERK5 pathway (14 genes), the JNK pathway (52 genes), the p38 MAPK pathway (45 genes) and the ERK1/2 pathway (73 genes) (Supplementary File 1).

We then calculated the somatic mutation frequencies in these 142 genes as determined in 192 cancer studies focusing on 101 different human cancer types and involving 40,848 patients (Supplementary File 1). The mutation frequencies for different genes ranged from 0.05% for the *CDC42* gene to 34% for the *TP53* gene. Other genes with high frequencies of mutations across the 40,848 patient samples were *KRAS* (10%), *BRAF* (6%) and *NF1* (5%; see Supplementary File 1 for the MAPK pathway gene mutations frequencies). These mutation frequencies are broadly consistent with previous reports involving smaller cohorts of ~10,000 TCGA tumours of a variety of different cancers, which indicated that, among receptor tyrosine kinase and ERK1/2 pathway genes, *KRAS* (9% across all samples) is the most frequently altered gene, followed by *BRAF* (7%)[17,28].

Although we found that the frequencies of MAPK pathway gene mutations are low when we considered the frequencies across all cancer types, we also found that the frequencies of mutations in some MAPK pathway genes were exceptionally high in some cancer types (Fig. 1, see Supplementary Fig. 1 for the complete connectivity of the MAPK proteins, also see Supplementary File 1). For instance, all esophagogastric cancer samples have *TP53* mutations, whereas 85% of pancreatic cancer samples have *KRAS* mutations, and 85% of the pilocytic astrocytoma samples have *BRAF* mutations. The oncogenes that were most frequently mutated in these tumours encode vital proteins that could be targeted to kill cancer cells selectively[29,30]. It is now known that despite the complexity of the mutational, epigenetic, and chromosomal aberration landscapes found across cancer cells, the survival of these cells remains dependent on the signalling functions of these frequently mutated MAPK pathway genes[29–31].

Overall, we found mutations in MAPK pathway genes in 58% of all tumours. Here, of the four major MAPK pathway modules, the JNK pathway (42.1% of the tumours) and the p38 pathway (40.3%) showed the highest frequencies of MAPK pathway gene mutations, followed by the ERK1/2 pathway (33.7%) and the ERK5 pathway (6.1%); (Fig. 1). The TP53 mutations accounted for over 28% of all the gene mutations in samples with JNK and p38 pathways mutations. By excluding TP53 mutations from our counts, we found JNK pathway mutations in 14.0% of all tumours and p38 pathway mutations in 11.3% of all tumours. We excluded the TP53 mutations from our downstream analyses because of the

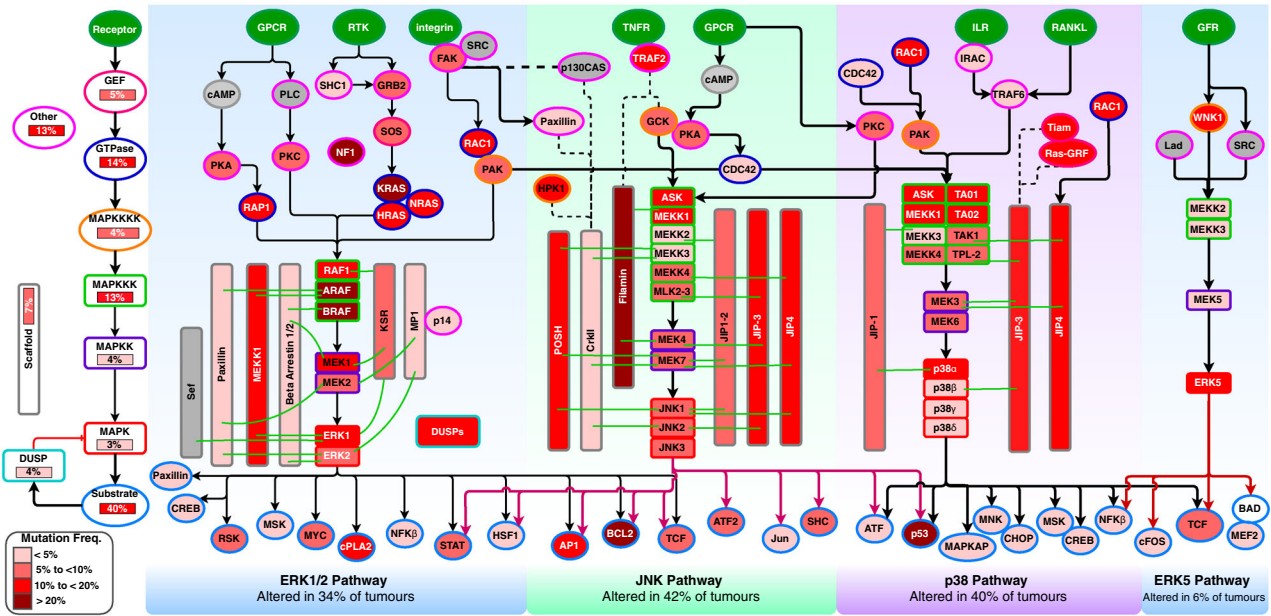

**Fig. 1 Mutations of MAPK pathway genes.** The nodes represent genes of the four MAPK pathway modules (the ERK1/2, p38, JNK and ERK5 pathways). The outline colours represent the classes of MAPK proteins that are encoded by particular genes, as shown on the left side of the figure. The abbreviations are as follows: GEFs guanosine exchange factors, GTPase Guanosine triphosphatase, MAPKKKK mitogen-activated protein kinase kinase kinase kinase, MAPKKK mitogen-activated protein kinase kinase kinase, MAPKK mitogen-activated protein kinase kinase, MAPK mitogen-activated protein kinase, DUSP dual-specificity phosphatase, Other; proteins other than those described above. Node colours represent the frequencies of gene mutations, and their increasing colour intensities denote higher percentages of tumours with mutations in genes encoding MAPK pathway proteins. In order clearly show the cancer-type mutations in the MAPK genes, we have presented the maximum gene alteration seen specific cancer types for each gene. This is because the mean gene mutation frequencies do not show how certain cancer types tend to have mutations in different MAPK genes. Additional, both the mean gene mutations across cancer types and the cancer-type maximum mutation rate in each specific MAPK pathway gene are given in Supplementary File 1 in the sheet named "Specific MAPK Gene Mutations". Edges indicate known types of interaction: red for inhibition, black arrows for activation, dotted lines for physical protein-protein interactions and green lines for interaction with scaffold proteins. To make the visualisation clearer, we have omitted some interactions between some network nodes. See Supplementary Fig. 1 for the complete connectivity network of all the MAPK pathway proteins.

high impact of TP53 mutations on the frequency of JNK and p38 pathway gene alterations. Furthermore, we found that 11% (4603 samples) of all the tumours harboured mutations in genes involved in more than one of the four MAPK pathway modules (Fig. 2a and Supplementary Fig. 2a).

Furthermore, we found mutations in genes that encode multiple classes of MAPK proteins in 15% (6481) of the patent samples (Fig. 2b). Among the genes that encode various classes of MAPK pathway proteins, the GTPase encoding genes (14.2%) showed the highest mutation frequencies, followed by the "other protein" encoding genes (13.2%) and the MAPKKK genes (12.7%; Fig. 1b).

Remarkably, we found that for most cancer types patient tumours tend to have high mutation frequencies in genes involved in either the ERK1/2 pathway or the p38 and JNK pathways, but rarely in the genes involved in both the ERK1/2 pathway and p38 and/or JNK pathways (Fig. 2c and Supplementary Fig. 2b).

Among the 101 cancer types that we analysed, we found that the extent to which the MAPK pathway genes were mutated varied (Supplementary File 1). The percentage of tumours with mutations ranged from 0% in small cell carcinomas of the ovary to 93% in pilocytic astrocytoma (Supplementary File 1).

**Disease outcomes are impacted by which MAPK pathways have mutated genes**. We examined whether the mutations in MAPK pathway genes regardless of any other covariates are associated with different clinical outcomes. Interestingly, we found that the median duration of overall survival (OS) for patients with mutations in MAPK pathway genes (OS = 73.0 months) was

significantly shorter ($p = 6.89 \times 10^{-8}$; log-rank test[32]) than that of patients with no mutations in MAPK pathway genes (OS = 81.2 months; Fig. 2d). However, we observed that disease-free survival (DFS) periods were similar (log-rank p-value = 0.455) for cancer patients with mutations in MAPK pathway genes (undefined median DFS period in that >50% of patients survived beyond the study duration) and those with no mutations in these genes (median DFS = 123.8 months; Fig. 2e). This observed impact of mutations in MAPK pathway genes on OS is consistent with previous ovarian, acute lymphoblastic leukaemia, and colorectal cancer studies which have reported that activation of the MAPK pathways is associated with worse clinical outcomes[33–36].

Next, we compared the duration of the OS and DFS periods between groups of patients with tumours that had: (1) mutations in genes of only one of the four MAPK pathway modules (i.e., the ERK1/2, ERK5, p38 or JNK pathways), (2) in genes of more than one of the MAPK pathway modules (3) or with no mutations in any MAPK pathway genes. We found that patients of these different subgroups exhibited dissimilar OS (log-rank $p = 1.51 \times 10^{-39}$; Fig. 2f) and similar DFS (log-rank $p = 0.0694$; Fig. 2g) period durations. We found that patients with tumours that had mutations in JNK pathway genes had the most favourable OS (median survival = 141.7 months) and DFS (undefined median DFS) outcomes, a finding that is consistent with other studies that have shown an association between alterations in JNK pathway genes and both enhanced apoptosis[37–39] and improved survival outcomes[37,38,40] (Fig. 2f, g). In contrast, patients with tumours that had mutations in genes of the ERK1/2 pathway exhibited the worst outcomes (median survival = 58.6 months; Fig. 2f). Furthermore, we found no significant difference in OS outcomes

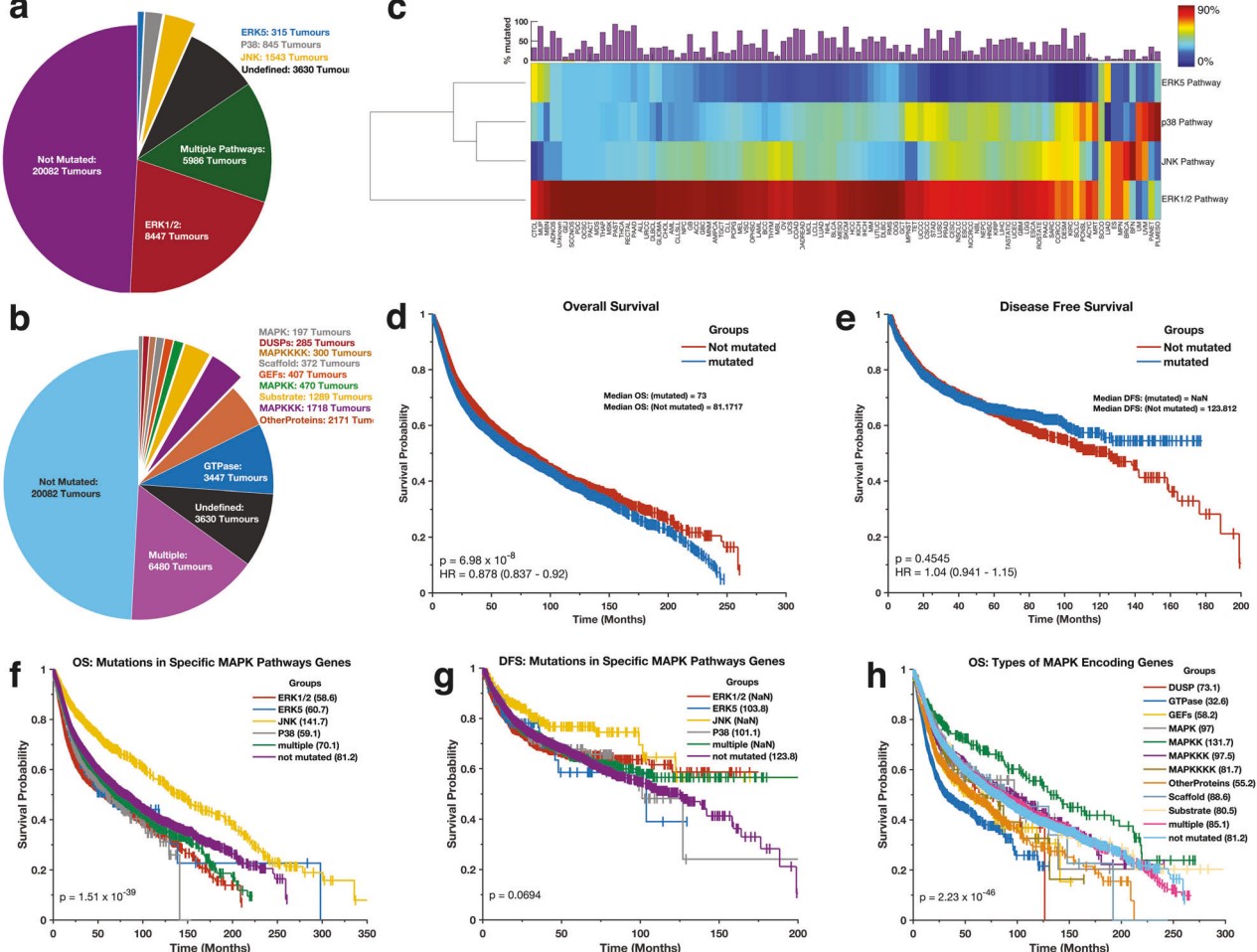

**Fig. 2 Mutations of the MAPK pathways across cancer types and disease outcomes. a** Pie chart indicating the proportions and the actual numbers of mutations within the genes of each MAPK pathway signalling module (ERK1/2, p38, JNK and ERK5 pathway). Note: the "undefined group" relates to tumour samples that were profiled by a targeted sequencing approach for which the sequencing panel included only some of the MAPK pathway genes that are known (1) oncogenes, and/or (2) tumour suppressor genes, and/or (3) frequently mutated in cancer (please refer to the Methods section for details). **b** Pie chart showing the proportions and the actual numbers of mutations to genes that encode different classes of MAPK proteins. **c** Clustering of the 101 distinct cancer types based on the proportions of samples with mutations in each of the four MAPK signalling pathway modules (Excluding TP53 mutations). Redder colour intensities denote higher percentages of mutations. The clustergram was produced using unsupervised hierarchical clustering with the cosine distance metric and complete linkage. The coloured bars on the heatmap show the overall frequency of gene mutations within the samples belonging to each cancer type represented within each column of the heatmap. Kaplan–Meier curve of the overall survival periods (**d**) and disease-free survival periods (**e**) of patients afflicted with tumours with and without mutations in MAPK pathway genes. Kaplan–Meier curve of the overall survival periods (**f**) and disease-free survival periods (**g**) of patients with tumours that have mutations to genes of only a single MAPK signalling pathway module (ERK1/2, p38, JNK and ERK5 pathway), mutations in genes of multiple MAPK signalling pathway modules, and no mutated MAPK pathway genes. The numbers in parenthesis show the median OS or DFS periods. NaN (Not a Number) represent undefined median OS or DFS period in that >50% of patients survived beyond the study duration. **h** Kaplan–Meier curve of the overall survival periods of patients with tumours that have mutations to genes that encode the various classes of MAPK proteins.

for patients with tumours that had mutations in the MAPK pathway module genes of the ERK1/2 pathway (median survival = 58.6 months) versus those of the ERK5 pathway (60.7 months; $p = 0.4$); the ERK1/2 pathways genes versus those of the p38 pathway (59.1 months; $p = 0.078$); and the ERK5 pathway genes versus those of the p38 pathway ($p = 0.74$; Fig. 2f; also see Supplementary File 2 for all pairwise comparisons).

Furthermore, we compared the durations of OS and DFS periods between groups of patients that had tumours with mutations in genes that encode specific protein classes involved in the MAPK pathways. Here, we found that patients segregated into these different protein class subgroups also exhibited distinctive durations of OS (log-rank $p = 2.23 \times 10^{-46}$; Fig. 2h) and DFS (log-rank $p = 3.83 \times 10^{-7}$; Supplementary Fig. 2c). For

the OS periods, patients with tumours that only had mutations in the GTPase class of genes (median survival = 32.6 months), exhibited the worst outcomes. In contrast, patients with tumours that had mutations in the MAPKK encoding genes (median survival = 131.7 months) exhibited the most favourable outcomes (Fig. 2h; also see Supplementary File 2 for all pairwise comparisons).

Our results, therefore, demonstrate an association in patient tumours between specific MAPK pathway gene mutations and disease aggressiveness.

**Cancer cells are dependent on MAPK pathway genes for their survival.** The purposeful disruption of genes in human cells

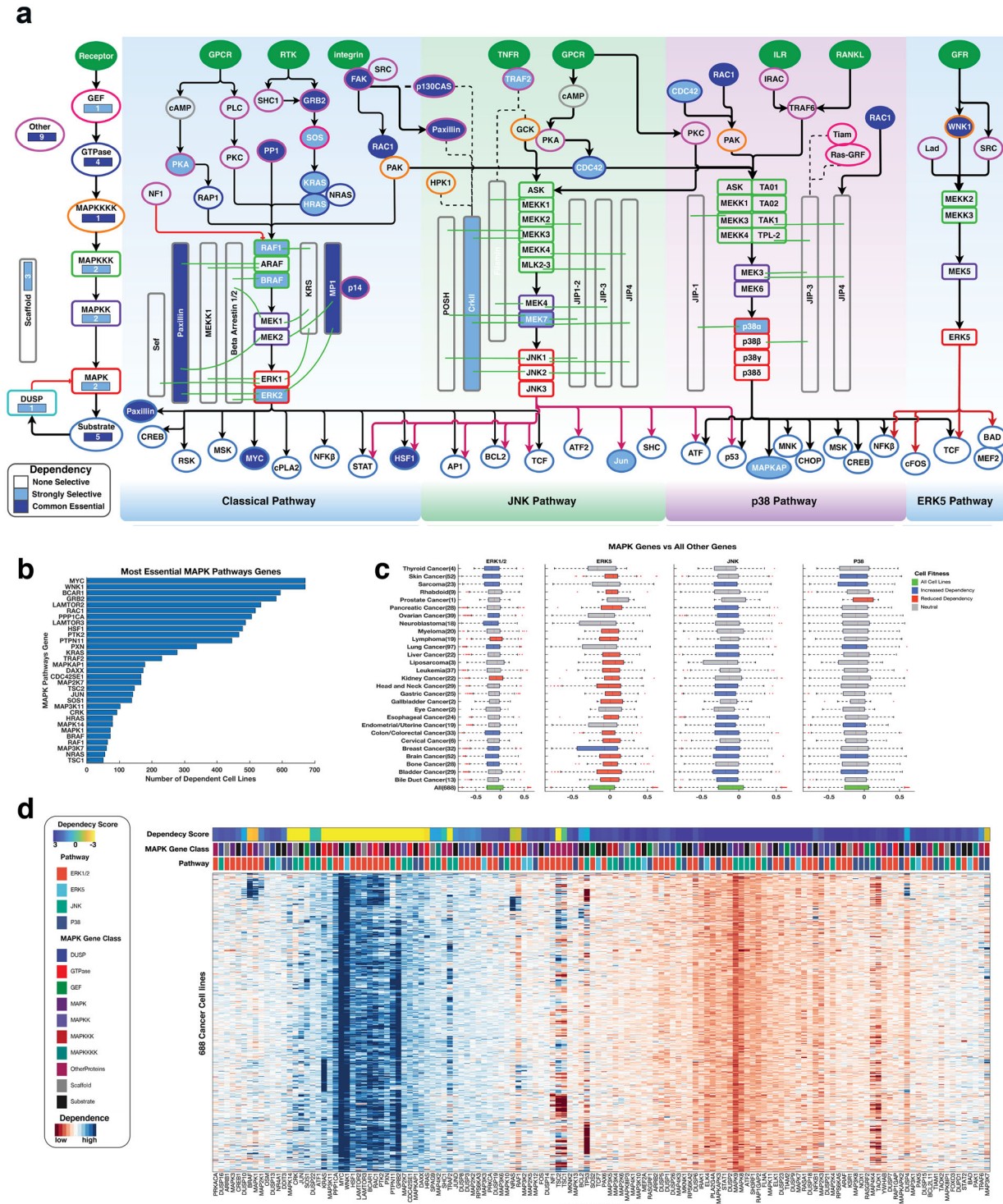

remains an integral approach to elucidating gene functions. Such an approach also holds great promise for finding therapeutic targets for combating diseases such as cancer. To identify genes that impact the fitness of cancer cells, we analysed the fitness impacts of CRISPR-based MAPK pathway gene knockouts carried out by the Achilles project[20] on 688 cancer cell lines drawn from 28 different tissues.

We found that cancer cell lines are significantly more dependent (i.e., they lose a higher degree of fitness after gene knockout) on having functional ERK1/2 pathway genes than they are dependent on having functional ERK5, p38, and JNK pathway

genes (Fig. 3a). We found eight MAPK pathway genes (including *MYC*, *WNK1* and *PXN*) that are classifiable as "common essential" (see the "Methods" section) among all cancer cell lines (Fig. 3a, b; Supplementary File 3). Recent studies show that, in malignant cells, the expression of the *MYC* oncogene is crucial for cancer cells to colonise organs at the expense of less performant neighbours. As a consequence of this, a functional *MYC* gene is necessary for the survival and clonal expansion of tumorous masses[41–43].

We also identified another 40 MAPK pathway genes (including *MAPK1*, *BRAF* and *MAPK14*) that are classifiable as "strongly

**Fig. 3 The dependence of cell lines on MAPK pathway genes. a** CRISPR-derived gene dependencies of the MAPK pathway genes. The nodes represent genes of the four MAPK pathway modules (the ERK1/2, p38, JNK and ERK5 pathways). Node colours represent the level of gene dependence, with increasing colour intensities denoting increasing gene dependencies. Edges indicate known types of interaction: red for inhibition, black arrows for activation and dotted lines for physical protein-protein interactions. Outline colours represent the classes of MAPK proteins that are encoded by genes, as shown on the left side of the figure. see Fig. 1 for their descriptions. **b** Bar graph showing the number of cell lines that are dependent on each gene for their fitness. Note: Fig. 3a shows the interacting MAPK protein names, whereas Fig. 3b shows the gene names. **c** Showing the comparisons of the mean dependence scores for each cancer vs the pooled mean dependence scores of all other MAPK pathway genes across all cancer types. From left to right, we show the dependence on the ERK1/2 pathway, JNK pathway, p38 pathway and ERK5 pathway of each cancer types compared to the pool dependence scores of non-MAPK pathway genes across all 688 of the cancer cell lines (the green coloured boxplots). The grey boxplots denote no difference, the blue boxplots denote loss of fitness, and the red boxplots denote increased fitness. *P*-values for each comparison was calculated with Welch's t-test. Within each box, the central mark denotes the median. The left and right edges of the box indicate the 25th and 75th percentiles, respectively. **d** An integrated plot of gene dependencies in cell lines. From top to bottom, the panel indicates the overall Achilles dependence scores of a single gene per column; the class of the MAPK pathway protein encoded by the gene; the MAPK pathway module in which the gene participates; and the clustergram of CRISPR-derived genes dependence scores across the cell lines. Blue colours indicate reduced fitness and red colours increased fitness after CRISPR-mediated gene knockouts. The clustergram was produced using unsupervised hierarchical clustering with the Euclidean distance metric and complete linkage (see Supplementary Fig. 3a).

selective" (see the "Methods" section) across various cancer cell lines. Here, unlike some genes which were found to be non-essential (i.e., genes that when knocked out do not affect the fitness of cells), all MAPK pathway genes impact the fitness of the 688 cell lines tested.

We found that overall, the fitness of the cell lines was more highly dependent (ANOVA *F*-value = 1365, $p < 1 \times 10^{-300}$) on the functionality of known oncogenes (both those of the MAPK pathway and those of other pathways) than on the functionality of known tumour suppressor genes (TSG; Supplementary Fig. 3a). Remarkably, we found that the fitness of these cell lines increased to a significantly greater degree when TSGs of the MAPK pathway were knocked-out compared to when any other known TSGs were knocked-out (Welch *t*-test, Bonferroni adjusted *p*-value = $5.4 \times 10^{-285}$; Supplementary Fig. 3a). Here, our findings emphasize that cellular processes such as cell proliferation, cell differentiation, cell survival, and cancer dissemination, which are all driven by the MAPK pathway, are essential determinants of cancer cell fitness[1–4,44,45].

Next, for cancer cell lines of various human cancer types, we compared the mean cell fitness dependency scores (derived using the CRISPR screens) between genes of each of the four MAPK pathway modules against all other genes expressed in these cells. Here, we found that ten cancer types (including those of the lung [Welch *t*-test, $t = 5.4$, $p = 7.7 \times 10^{-8}$], skin [$t = 9.8$, $p = 1.14 \times 10^{-21}$] and brain [$t = 7.3$, $p = 3.35 \times 10^{-13}$]) were significantly more dependent on signalling through the ERK1/2 pathway than were other cancer cell types (Fig. 3c; Supplementary File 3). Surprisingly, we found that when the oncogenes of the ERK5 pathway were knocked out, 19 of the 28 cancer types that are represented in the Achilles datasets tended to have increased fitness (Fig. 3c). Also, we showed that certain cancer types have a higher dependence on signalling through various MAPK pathway modules compared to other cancer types (Supplementary Fig. 4).

For cancer cell lines of various human cancer types, we compared the mean cell fitness dependency scores between genes that encode different classes of MAPK pathway proteins, against all other genes expressed in these cells. Here, we found that 21 cancer types (including pancreatic cancer [Welch *t*-test, $t = 6.4$, $p = 1.1 \times 10^{-9}$], bladder cancer [$t = 6.5$, $p = 7.2 \times 10^{-10}$] and colorectal cancer [$t = 6.9$, $p = 4.2 \times 10^{-11}$]) are significantly more dependent on GTPase encoding genes than they are in other classes of genes (Supplementary Fig. 5a, b; Supplementary File 3).

Finally, by clustering the CRISPR-mediated gene knockout data of the cell lines, we revealed a group of ERK1/2 pathway genes that all adversely affect the fitness of cell lines in an analogous manner (Fig. 3d; Supplementary Fig. 3b).

**Gene essentialities are correlated with gene transcriptional signatures.** We compared the relationships between CRISPR-determined gene dependency data and mRNA transcription, DNA methylation and CNV data. Here, we found that the gene essentiality signature of the cell lines is related to their mRNA transcription signature (Fig. 4a). Therefore, we assessed the correlation between the mRNA expression levels of all MAPK pathway genes and their CRISPR-derived dependency scores and uncovered a statistically significant negative correlation ($R = -0.32$, $p < 1 \times 10^{-300}$; Fig. 4b). We also revealed that *MYC*, a MAPK pathway gene on which most of the tested cell lines (671 out of 678) are dependent for their fitness, has self-mRNA transcript levels that are negatively correlated across the tested cell lines with CRISPR-derived dependency scores (Fig. 4c). This suggests that MYC-driven endogenous cell competition likely results in cells with higher *MYC* levels being selectively preserved due to the apoptotic elimination of cells with lower *MYC* levels[41,42,46].

Since we found an overall negative correlation between the mRNA expression levels and the gene dependence scores of the MAPK pathway oncogenes, we hypothesised that the "common essential" genes are likely to be highly expressed in cancer cell lines. Therefore, we compared the mean transcript levels between the "common essential" MAPK pathway genes and other MAPK pathway genes and found that the "common essential" genes are indeed significantly more highly expressed (Welch *t*-test; $t = 99$; $p < 1 \times 10^{-300}$; Fig. 4d). This suggests to us that the high mRNA transcript levels and the negative correlation between these levels and CRISPR-derived dependency scores for the MAPK pathway genes may represent a form of selective gene expression amplification phenomenon similar to that displayed by *MYC* during oncogenesis[43,46–48].

Given that, among all of the MAPK pathway genes, *KRAS* and *BRAF* were the most frequently mutated oncogenes, we focused on the impacts of these two genes on cell fitness. We found that pancreatic cancer cell lines, 85% of which present with *KRAS* mutations, were significantly more dependent on *KRAS* than were all other cell lines ($t = 35.4$, $p = 2.8 \times 10^{-178}$; Fig. 4e). Furthermore, we found that cell lines with *KRAS* mutations were significantly more dependent on *KRAS* expression than were all other cell lines ($t = 34.7$, $p = 1.4 \times 10^{-187}$). Here, also, we found that skin cancer cell lines were significantly more dependent on *MAPK1* ($t = 16.5$, $p = 1.33 \times 10^{-55}$; Fig. 4f) and *BRAF* ($t = 17.9$, $p = 1.6 \times 10^{-63}$; Fig. 4g) expression than were other cell lines. Furthermore, we found that the cell lines with *BRAF* mutations are more dependent on *BRAF* expression than were all other cell lines ($t = 19.5$, $p = 8.3 \times 10^{-75}$; Fig. 4g). Overall, we observed that

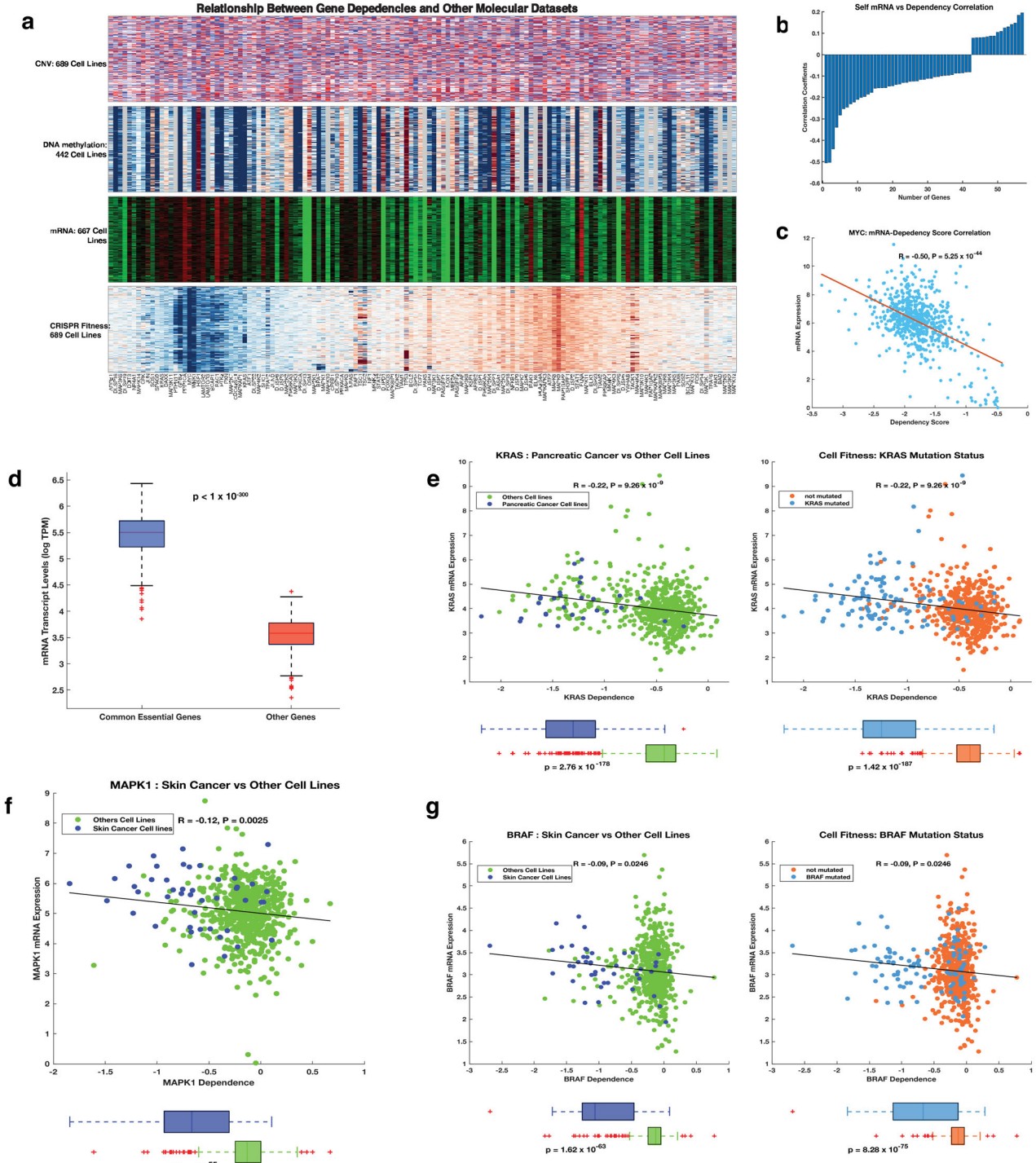

**Fig. 4 Relationship between Achille gene dependence scores and other molecular profiling datasets. a** Integrated plot depicting the relationship between CRISPR-derived gene dependence scores and, from top to bottom, gene copy number variations; the DNA methylation profiles; the mRNA transcription levels; and gene dependence scores. **b** Bar graph showing the statistically significant Pearson's correlation scores between the mRNA expression levels of oncogenes encoding MAPK pathway proteins and their corresponding gene dependence scores. **c** Pearson's correlation between *MYC* proto-oncogene mRNA expression levels and the CRISPR-derived *MYC* gene dependence score. **d** Comparison of mRNA transcript levels. **e** From left to right: correlation between the *KRAS* transcript levels and the *KRAS* gene dependence scores, and the mean difference in the *KRAS* dependence between pancreatic cancer cell lines (pancreatic cancers have the highest frequencies of *KRAS* mutations) and all other cancer cell lines; the mean difference in *KRAS* dependence score between *KRAS* mutant cell lines and cell lines that do not harbour *KRAS* mutations. **f** Correlation between the *MAPK1* transcript levels and the *MAPK1* dependence scores, and the mean difference in *MAPK1* dependence score between skin cancer cell line and all other cancer cell lines. **g** (Left) correlation between *BRAF* transcript levels and *BRAF* dependence scores, and the mean difference in the *BRAF* dependence score between skin cancer cell lines (cancer type with most *BRAF* mutations) and all other cancer cell lines. (Right) The mean difference in the *BRAF* dependence score between *BRAF* mutant cell lines and cell lines that do not harbour *BRAF* mutations.

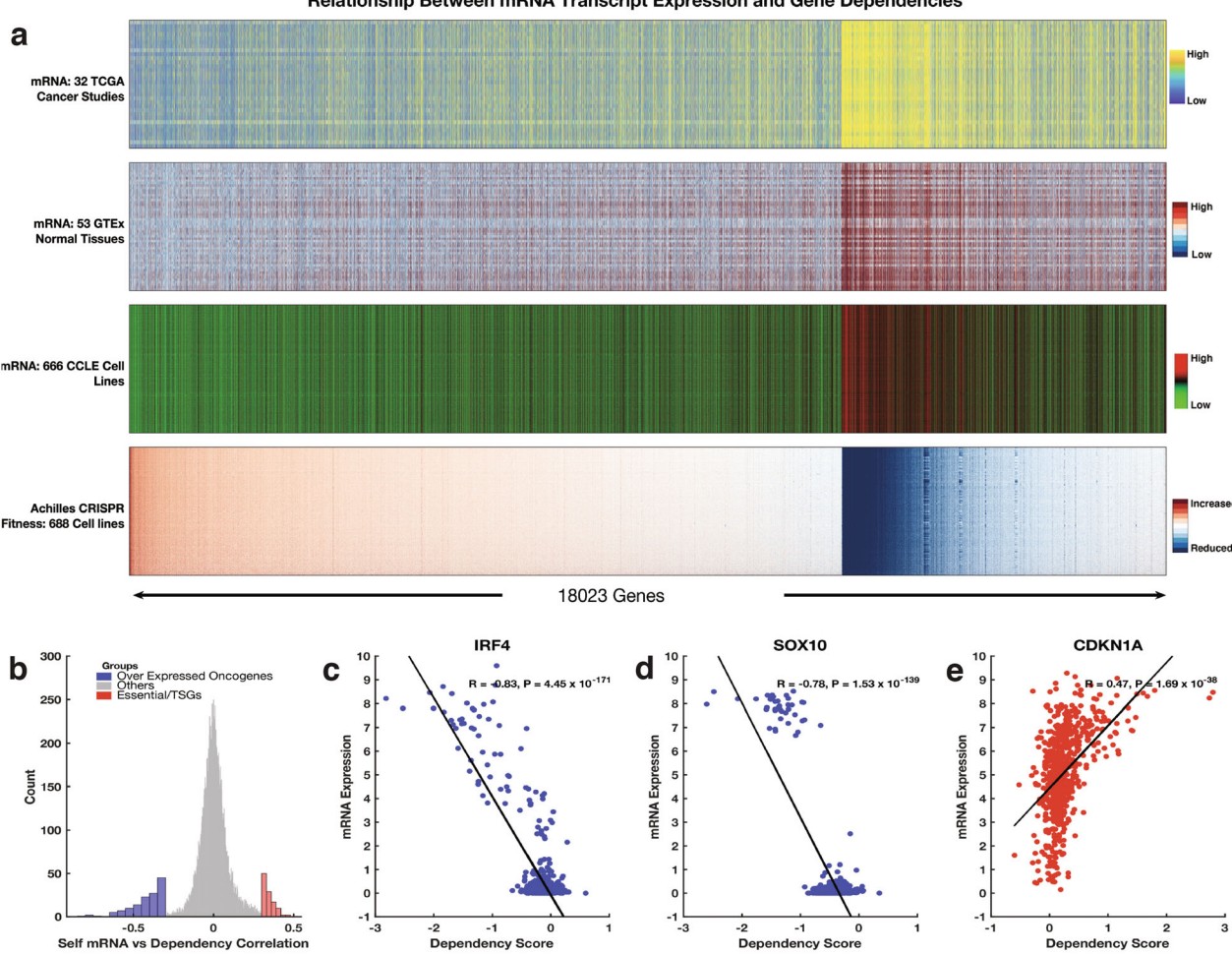

**Fig. 5 Relationship between Achille gene dependence scores and the mRNA transcriptional abundance of all the protein-coding genes. a** Integrated plot showing the relationship between CRISPR-derived gene dependence scores for over 18,000 protein-coding genes and the mRNA transcript levels of these genes. From top to bottom: mRNA transcript levels of 10,534 primary tumour samples profile by the TCGA pan-cancer project; mRNA transcript levels of 53 non-diseased tissue sites measured in nearly 1000 individuals by the GTEx consortium; mRNA transcript levels of 667 cell lines profiled by the CCLE; Achilles CRISPR knockout fitness screens of 688 cell lines. **b** Histogram showing the overall correlation between gene dependence and the gene's self-mRNA levels. To aid visualisation, we used a bin size of 300 for genes (shown as "other genes") that showed a Pearson's correlation coefficient between −0.3 and 0.3. In contrast, we used a bin size of 10 for the genes that showed a Pearson's correlation coefficient of <−0.3 and or >0.3. Correlation between gene dependence and the gene's self-mRNA levels for **c** *IRF4* transcription factor, **d** *SOX10* transcription factor, and **e** the *CDKN1A* genes.

CRISPR-mediated disruptions of *KRAS*, *BRAF*, and *NRAS*, was associated with some of the most robust decreases in cellular fitness. However, *KRAS* ($R = −0.22$, $p = 9.3 \times 10^{−9}$), *BRAF* ($R = −0.09$, $p = 0.03$), and *NRAS* ($R = −0.14$, $p = 0.0003$) also displayed weak negative correlations between self-mRNA levels and their CRISPR-derived gene dependency scores (Fig. 4e, g, and Supplementary Figure 3c). This suggests that in particular cancers, for *KRAS*, *BRAF*, and *NRAS*, the major drivers of oncogenesis are gene mutations rather than cellular mRNA levels.

Since the correlation between gene expression signatures and CRISPR-derived gene dependency scores was unexpected, for the 18,023 genes that had corresponding CRISPR-derived gene dependency data, we evaluated the relationship between the mRNA transcript levels and gene dependency scores. Here, using mRNA transcription data of cell lines from the CCLE, primary cancer tissues profiled by the TCGA, and normal tissues profiled by the GTEx consortium[49], we found an association across different tissues between mRNA expression levels and the degree to which CRISPR-mediated inactivation of all the 18,023 genes impacted the fitness of cell lines (Fig. 5a; Supplementary Fig. 6a).

We found that various oncogenes and transcription factors display a negative correlation, whereas TSGs show a positive correlation between CRISPR-derived gene dependency scores and self-mRNA levels (Fig. 5b; Supplementary File 4). Here, the oncogenes and transcription factor genes that showed linear correlations with associated Pearson's correlation coefficient values < −0.3 were enriched for, among others, biological processes associated with regulation of transcription from RNA polymerase II promoters and regulation of cell proliferation (Supplementary Fig. 6b, Supplementary File 3). Alternatively, the genes that showed linear correlations with an associated Pearson's correlation coefficient values > 0.3 were enriched for, among others, biological processes that are associated with negative regulation of the mitotic cell cycle phase transition and regulation of G2/M transition of the mitotic cell cycle (Supplementary Fig. 6c, Supplementary File 3).

Among the known oncogenes and transcriptions factors, *IRF4* ($R = −0.83$, $p = 4.5 \times 10^{−171}$; Fig. 5c) showed the strongest negative correlation, followed by *SOX10* ($R = −0.78$, $p = 1.5 \times 10^{−139}$; Fig. 5d). Conversely, *CDKN1A* ($R = 0.47$, $p = 1.7 \times 10^{−38}$) showed the strongest positive correlation among the

known TSGs (Fig. 5e; Supplementary File 4). These results are entirely consistent with our current understanding of cancer cell biology in that we expect the disruption of an oncogene to reduce the fitness of cancer cells, whereas we expect the disruption of a TSG to increase the fitness of these cells[50–52].

Altogether, these analyses revealed a relationship between the extent of mRNA transcription and/or mutations in MAPK pathway genes and the dependence of cancer cells on these potentially dysregulated genes for survival.

The drug responses of cell lines to MAPK pathway inhibitors are associated with the degree to which they depend on targeted MAPK pathway components.

Given the association between the degree to which the fitness of cancer cell lines depended on the functionality of different MAPK pathway genes and the degree to which those genes were expressed in these cell lines, we were interested in determining whether CRISPR-derived measures of how dependent cells were on MAPK pathway components correlated with cellular responses to existing drug molecules that inhibited these components. If such a correlation existed, it would mean that CRISPR-derived measures of MAPK pathway gene dependence would provide predictive power for identifying the best drug targets within the MAPK pathways. Therefore, we retrieved, from the GDSC database, the dose-responses of 344 cancer cell lines representing 24 different human cancer types to 28 MAPK pathway inhibitors (Supplementary File 4)[21].

For each of the MAPK pathway genes, we grouped the cell lines into two groups: those with a high degree of dependence on that particular gene and those with low dependence on that gene (see the "Methods" section). We then compared the mean dose-response of the two cell-line groups using the drug response data from the GDSC (see Fig. 6a). Further, we grouped the cell lines into another two groups: those with mutations in a particular MAPK pathway gene and those without mutations in that gene (see the "Methods" section). Again, we then compared the dose-response of the two groups using drug response data from the GDSC (Fig. 6a and Supplementary Fig. 7a). Altogether these two sets of comparisons revealed two critical insights that are of relevance to the use of MAPK pathway inhibitors as cancer therapeutics.

The first insight was that the responses of cell lines to MAPK pathway inhibitors did indeed vary with the degree to which the cell lines were dependent on particular MAPK pathway genes. Specifically, we found that CRISPR-derived dependence scores of 15 genes were associated with significantly increased sensitivity of cell lines to the MAPK pathway inhibitors, and the dependence scores of twelve genes were associated with significantly decreased sensitivity to these inhibitors (Fig. 6a). We also found that the CRISPR-derived dependence scores of 18 other genes were associated with mixed responses; i.e., significantly increased sensitivity to some of the inhibitors and significantly decreased sensitivity to others (Fig. 6a). Here also, for cell lines with higher CRISPR-derived dependence scores to a specific MAPK pathway gene, dependency on *TSC1* was associated with significantly increased sensitivity to the most (64%) MAPK pathway inhibitors, followed by dependence on *MAPK1* (48%) *BRAF* (48%), and *NRAS* (46%; see Supplementary File 4). Conversely, across the cell lines, dependence on *DUPS8* was associated with significantly reduced sensitivity to 28% of the inhibitors, followed by dependence on *TRAF6* (16%) and *DUSP13* (8%), *DUSP19* (8%), *MAP2K3* (8%) and *MAP3K3* (8%; see Supplementary File 4).

The second insight was that the response of cell lines to MAPK pathway inhibitors is also related to the specific mutations that the cell lines carry in MAPK pathway genes. We found that mutations in 41 genes were associated with significantly increased sensitivity of cell lines to the MAPK pathway inhibitors, and 29 genes were associated with significantly decreased sensitivity (Fig. 6a and Supplementary Fig. 7a). Here, mutations in another 24 genes were associated with mixed responses of the cell lines to the MAPK pathway inhibitors, i.e., significantly increased sensitivity to some of these inhibitors and significantly decreased sensitivity to others (Fig. 6a and Supplementary Fig. 7a). Furthermore, for cell lines with mutations in specific MAPK pathway genes, mutations in the *BRAF* gene were associated with significantly increased sensitivity to the most MAPK pathway inhibitors (44%), followed by mutations in *NRAS* (20%), *MAP3K5* (20%), *CRK* (20%) and *RASGRF1* (20%; see Supplementary Fig. 7a and Supplementary File 4). Conversely, across the cell lines, mutations in *BCL2L11* were associated with significantly decreased sensitivity to 24% of the inhibitors, followed by mutations in *TRAF6* (12%), *PRKACA* (12%), and *PTPN11* (12%; see Supplementary File 4).

Altogether, we found 543 instances where either a high dependence on MAPK pathway genes, or mutations in MAPK pathway genes, were significantly associated with variations in the dose-responses of cancer cell lines to anticancer drugs (Fig. 6b). Among the cell lines that are highly dependent on MAPK pathway genes, we found 237 instances where the cell lines were significantly more sensitive to MAPK pathway inhibitors and 143 instances where the cell lines were significantly more resistant to these inhibitors (Fig. 6b). Furthermore, among the cell lines that have mutations to specific MAPK pathway genes, we found 100 instances where cell lines were significantly more sensitive to MAPK pathway inhibitors and 63 instances where cell lines were significantly more resistant to these inhibitors (Fig. 6b). This then indicates that a high degree of dependence on MAPK pathway genes (480 total instances) influences the anticancer drug responses of assessed cell lines to a greater degree than do mutations within these cell lines (163 total instances).

Next, we classified the cancer cell lines into another two categories; those with either a generally higher or lower CRISPR-derived dependency on MAPK signalling (see the "Methods" section). We then compared drug IC50 values between these two cell line groups for the 28 MAPK pathway inhibitors. Remarkably, we found that the cell lines with higher MAPK gene dependency were significantly more sensitive to eight out of the 28 MAPK pathway inhibitors (Supplementary Fig. 7b; Supplementary File 4). The inhibitors that exhibited the most significant difference in their dose-responses between the two cell line groups were refametinib ($t = 5.5$, $p = 6.1 \times 10^{-08}$), trametinib ($t = 4.5$, $p = 9.7 \times 10^{-06}$) and selumetinib ($t = 4.5$, $p = 9.9 \times 10^{-06}$), all of which target MEK1 and MEK2 (Supplementary Fig. 7b; Supplementary File 4). The cancer cell lines that have either a higher or lower dependency on MAPK signalling are given Supplementary File 4.

Finally, we classified the 25-cancer types represented within the GDSC database into two categories: one with a higher CRISPR-derived dependency on MAPK genes and the other with lower dependency on these genes (see the "Methods" section). Again, we compared the mean dose-responses of the 28 MAPK pathway inhibitors between these two groups of cancer types. Here, we found that 14 MAPK pathway inhibitors were significantly more effective at killing cancers that had higher dependencies on MAPK pathway genes than they were at killing cancers with lower dependencies on these genes ($t = 5.9$, $p = 1.5 \times 10^{-51}$; Supplementary Fig. 7c). The MAPK inhibitors that exhibited the most significant differences in their efficacy between these two groups of cancer types were PD0325901 ($t = 5.6$, $p = 7.6 \times 10^{-09}$), selumetinib ($t = 6.0$, $p = 2.9 \times 10^{-08}$) and AZ628 ($t = 5.2$, $p = 6.9 \times 10^{-08}$; Fig. 6c). Here, the MEK1/MEK2 inhibitors all ranked in the top six (five out of the top six ranking inhibitors),

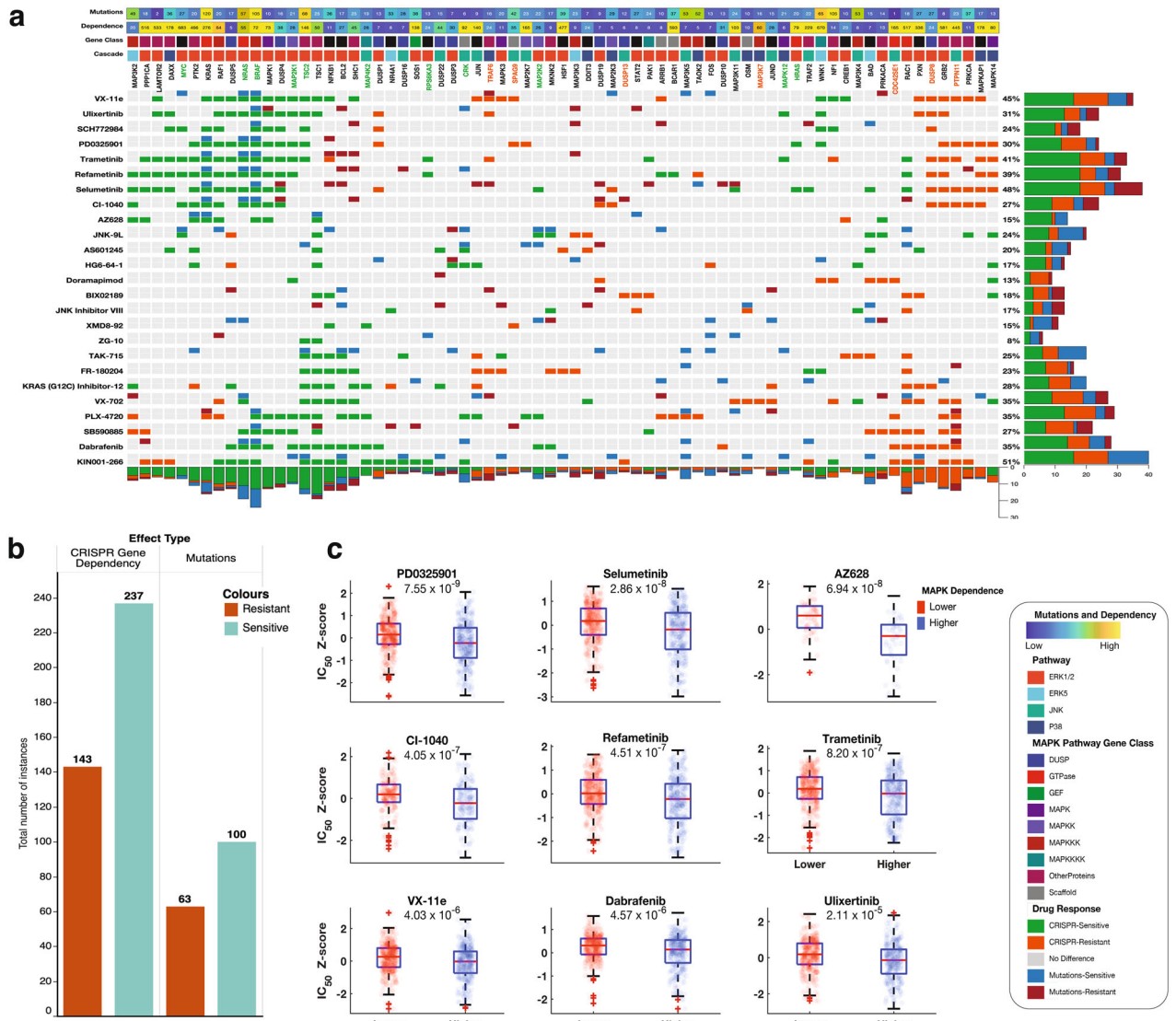

**Fig. 6 Relationship between Achille gene dependence scores or mutations of MAPK pathway genes and the responses of the cell line to MAPK pathway inhibitors. a** The integrated plot is showing the relationship between gene dependencies or mutations and drug responses across cancer cell lines. From top to bottom panels indicate: the overall mutation frequencies observed for the gene along that column. Dependence; the overall CRISPR-derived gene dependence scores of the gene along that column. Gene Class; the class of the MAPK pathway protein encoded by the gene. Pathway Module; the MAPK pathway module in which the gene participates. Clustered heatmap; The marks on the heatmap are coloured based on how a high dependence on, or mutations in, the gene along each column affect the efficacy of the drug given along each row: (1) with green (for gene dependence) and blue (for gene mutations) denoting significantly (10% false discovery rate) increased sensitivity, (2) grey for no statistically significant difference between cell line with a higher and lower dependence on the gene, and (3) orange (for gene dependence) and deep red (for gene mutations) denoting significantly increased resistance. The gene names (column labels) are coloured based on the overall calculated effect that high dependence on the gene has on the efficacy of the drug given along rows. Green; all the cell lines are significantly more sensitive to all the MAPK pathway inhibitors, orange; all the cell lines are significantly more resistant to all the MAPK pathway inhibitors, and black; a mixed response to MAPK pathway inhibitors. The bar graphs represent the total numbers of drugs whose dose-response are significantly increased (green/blue) or decreased (orange/deep red). **b** The total number of instances (from the heatmap) for which cell lines are either significantly more resistant or more sensitive to MAPK pathway inhibitors broken down into the effect types (i.e., CRISPR dependency scores or gene mutations). **c** Comparison of the dose-response profiles to MAPK inhibitors between the cancer types with higher dependence (boxplots with blue scatter point) on MAPK signalling and those with lower dependence (boxplots with red scatter point) on MAPK signalling. The cancer types that have either a higher or lower dependency on MAPK signalling are given Supplementary File 4. Boxplots show the logarithm transformed mean IC50 values of the cancer cell lines of each group. On each box, the central red mark indicates the median, and the bottom edge represents the 25th percentiles, whereas the top edge of the box represents 75th percentiles. The whiskers extend to the most extreme data points not considered outliers, and the outliers are plotted individually using the '+' symbol. The scatter point within each box plot shows the overall distribution of the data points.

revealing that cancers that were more dependent on functional MAPK pathway genes were likely to exhibit stronger responses to MEK1/MEK2 inhibitors than cancers that were less dependent on functional MAPK pathway genes. Here, none of the cancer types with a lower degree of dependency on MAPK pathway genes displayed significantly more sensitive to any of the MAPK pathway inhibitors than cancer types with higher degrees of dependency on MAPK pathway genes (Fig. 6c; Supplementary File 4). Moreover, previous studies have shown an association between drug sensitivity and both the expression levels of targeted proteins and/or the presence of alterations within these proteins[21–23,53–55]. Here, we also show a clear correlation between the gene dependencies and drug sensitivities of cancer cell lines.

Overall, these discoveries emphasise that the extent to which different cell lines or different cancer types are dependent on MAPK signalling defines their responsiveness to drugs that target the components of the MAPK pathway. This implies that CRISPR-derived estimates of the degree to which different MAPK pathway components contribute to cellular fitness are clinically relevant predictors of how different primary tumour types will respond to different MAPK pathway inhibitors.

**Transcription responses of the MAPK pathway genes to MAPK pathway inhibitors**. Since we found that the responses of cell lines to MAPK pathway inhibitors is related to their dependence on the functionality of specific MAPK pathway genes, we hypothesised that it should be possible to determine the exact cellular changes that are associated with drug responses. Here, the mRNA transcription patterns displayed by cells following their exposure to MAPK pathway inhibitors should provide a clear representation of these cellular changes. To evaluate this hypothesis, we used the publicly accessible LINCS dataset which details the mRNA transcription patterns of ~1000 genes in cell lines[25,26] following exposure of the cell lines to, amongst other small molecules, seven MAPK pathway inhibitors. Here we focused on the two cell lines (MCF7 and A549) and two MAPK pathway inhibitors (selumetinib and PD-0325901; both of which target mitogen-activated protein kinase kinase), that are common between the LINCS datasets, and the GDSC or CCLE datasets.

We retrieved the dose-response profiles from the CCLE for MCF7 and A549, to reveal that A549 ($IC_{50} = -1.481$ μm) exhibits a greater degree of sensitivity than MCF7 ($IC_{50} = 8.0$ μm) to PD-0325901 ($IC_{50} = 3.28$; Supplementary Fig. 7d). Similarity, the dose-response profiles from the GDSC for MCF7 and A549, revealed that A549 ($IC_{50} = 0.171$ μm) is more sensitive than MCF7 ($IC_{50} = 8.0$ μm) to selumetinib (Supplementary Fig. 7e).

Here, we found that changes in transcription in response to selumetinib and PD-0325901 were positively correlated for both A549 ($R = 0.99$, $p < 1 \times 10^{-300}$) and MCF7 ($R = 0.92$, $p < 1 \times 10^{-300}$) cells (Fig. 7a). Surprisingly, we found that the mRNA transcription signatures after treatment with selumetinib, and PD-0325901 were also highly correlated when comparing the cell lines to one another, despite A548 being substantially more sensitive to these drugs than was MCF7 (Fig. 7b).

To understand this paradox, we subtracted the transcription profile of the DMSO treated control from the transcription profiles of the two cell lines following their treatment with selumetinib or PD-0325901. Here, for A549 treated with either selumetinib or PD-0325901, we found expression changes to many genes, including a reduction in the mRNA levels of the genes, *MYC* and *WNK1* (Fig. 7c) which are defined by the Achilles project as being "common essential" and the genes *KRAS*, *HRAS*, and *MAPKAPK2* which are defined by the Achilles project as being "strongly selective" (Fig. 7c). Conversely, for

MCF7 treated with either selumetinib or PD-0325901, we observed increased mRNA levels of *WNK1* and *MAPKAPK2*, whereas the mRNA levels of the *NRAS* and *KRAS* genes were unchanged (Fig. 7c).

Our findings here are consistent with previous studies which have shown that differences in the survival of different tumour cells after drug perturbation can be at least partially explained by differences between the transcriptional signatures of the tumour cells[20,56]. What we have shown is that, in the context of cancer cell lines responding to MAPK pathway inhibitors (and probably also in the primary tumours from which these cell lines were derived), such sensitivity differences are very likely attributable to the extent to which the inhibitors reduce the expression and/or functioning of key cell fitness associated MAPK pathway genes.

## Discussion

We have conducted the most comprehensive analysis of the MAPK pathways in 101 different human cancer types. Here, we found at least one non-synonymous mutation to genes involved directly in MAPK pathways in 42% (58% when TP53 mutations are included) of all analysed samples. Previous studies have examined the frequencies of alterations to various other categories of genes in tumours including those involved in metabolic pathways (occurring in 100% of all tumours examined), the transforming growth factor pathway (39%), the PI3K-mTOR pathway (33%), the cell cycle pathway (33%), the p53 pathway (29%), the *MYC* oncogene and its proximal network (28%), and the Hippo pathway (10%)[17,28,57,58]. We have, therefore revealed that, after metabolic gene alterations, alterations of MAPK pathway genes are the most frequently observed category of genetic changes associated with the onset of human cancers.

Our findings emphasize the significance of the MAPK pathways across most cancer types in promoting and coordinating the proliferative capacity and immortality of cancer cells. Despite the significance during oncogenesis of alterations in MAPK pathway genes, it must be stressed that MAPK pathway gene alterations are not found in 42% of tumours. These mutations are infrequent in cancers such as small cell carcinoma of the ovary (occurring in none of the 15 examined tumours), Ewing sarcoma (occurring in 3% of the 122 examined tumours) and myeloproliferative neoplasms (occurring in 5% of the 151 examined tumours). These exceptions reinforce the notion that there exist multiple MAPK pathway independent routes to oncogenesis[59–63].

We showed that, on average, patients with tumours that harbour mutations in MAPK pathway genes tend to have significantly worse OS outcomes than those without such mutations. Since the MAPK pathway promotes tumour cell growth, proliferation, resistance to drug therapy, and tissue invasion, it is unsurprising that hyperactivation of this pathway can lead to more aggressive disease[1–4,6,8,14,64]. We find, however, that this is likely only the case for three of the four main MAPK pathway modules in that patients with tumours that have mutations in only JNK pathway genes on average tended to exhibit significantly better disease outcomes (as adjudged by both OS and DFS) than patients with tumours that have mutations in other MAPK pathway modules (Fig. 2g). Given that the activation of JNK signalling has a tumour suppressor role[65–67] in many contexts (but a tumour promotor role in others contexts[66–69]), the apparently enhanced survival of patients with tumours that have JNK pathway gene mutations suggests that, whenever these mutations have any impact on cancers at all, they may most commonly promote anti-tumour activity.

We found a link between the most frequently mutated oncogenes in various cancer types, e.g., *KRAS* mutations in pancreatic cancer (Fig. 4e) and *BRAF* mutations in skin cancer (Fig. 4g) and

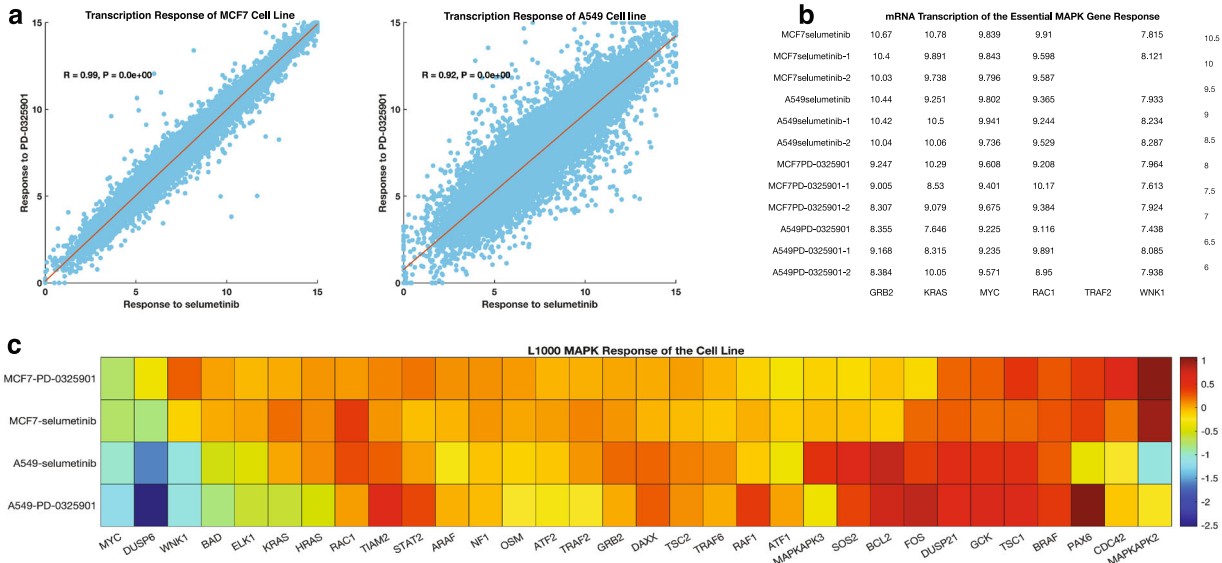

**Fig. 7 Transcription responses of the MAPK pathway genes to MAPK pathway inhibitors. a** Correlation between mRNA transcription response after treatment with PD-0325901 and selumetinib for the (left) A549 cell lines and (right) the MCF7 cell lines. **b** Heatmap showing the transcriptome changes of the A549 and MCF7 cell lines that occur after treatment of each cell line with either PD-0325901 or selumetinib for each repeat experiment done by the LINCS project. **c** Clustered heatmap showing the transcription response of the "common essential" and "strongly selective" MAPK pathway genes after-treatment of the A549 and MCF7 cell lines with either PD-0325901 or selumetinib.

the degree to which cancer cells depended on functional versions of these genes for survival. Some of these dependencies (so-called "Achille's heels") were noted before the era of large-scale CRISPR-based gene editing and are already either targeted by established chemotherapeutics or are being evaluated as targets for future therapeutics[70–73]. Therefore, we suggest that the CRISPR-based gene dependency screen performed by the Achilles project could be leveraged to identify a host of other drug targets.

We also showed that cancer cells with mutations in particular MAPK pathway genes respond more favourably to MAPK pathway inhibitors than do those without mutations into these genes (Fig. 6). Similarly, cell lines with a high degree of dependency on MAPK pathway genes also exhibit better responses to MAPK pathway inhibitors (Fig. 6) than cell lines that have a lower degree of dependency on these genes. Here, just as others have shown[23,53,54,56], our findings have linked gene dependencies and gene mutations to drug action. This underscores the notion that we could devise better treatment strategies for many human cancers by simply examining their: (1) MAPK pathway mutational landscapes; (2) the mRNA expression levels of vital oncogenic drivers; and (3) the degrees of the dependence of cancer cells on these oncogenes. Here, our speculation is further corroborated by our discovery, using the LINCS project datasets, that reduced transcription of crucial oncogenes and/or transcription factor genes are somewhat predictive of whether cancer cells will be sensitive or refractory to a particular anti-cancer drug. It should be interesting to unravel the signalling pathway mechanisms that have knock-down effects on the expression of genes that the Achilles project has identified as "common essential" or "strongly selective" since these mechanisms would also impact the responses of cancer cells to drug perturbation.

Altogether, we have revealed both the extent of mutations in the MAPK pathway genes across more than 100 human cancer types and the subset of these mutations that are most likely to impact disease outcomes. Our integrative analysis of the CRISPR-derived gene dependencies of cancer cell lines, together with the drug responses of these same cell lines, indicates that the mutations in, and expression signatures of, MAPK pathway genes are

associated with the responses of the cell lines to various MAPK pathway inhibitors. It is apparent; therefore, that it should be relatively straightforward to extend such an integrative analysis approach to identify high-confidence drug targets for a broad array of human cancers.

## Methods
We analysed a dataset of 40,848 patient-derived tumours representing 101 distinct human cancers, obtained from cBioPortal[19] version 3.1.9 (http://www.cbioportal.org; see Supplementary File 1 for details on the cancer studies). The elements of the data that we obtained from cBioPortal include somatic gene mutations (point mutations and small insertions/deletions), mRNA expression, and comprehensively deidentified clinical data.

Not all types of data were available for all patients because of assay failures, incomplete specimen availability and issues of quality with certain samples. Furthermore, not all the MAPK genes were sequenced in all samples because some 15 of the 192 cancer studies were profiled using targeted sequencing. Here, all statistics and results that we present are based on the subset of samples that have complete data for each MAPK pathway gene, the genes of each MAPK pathway module, or were applicable with at least a mutation within genes of a MAPK pathway module.

**The mutational landscape of the MAPK pathway genes.** Using the literature and the KEGG pathways database[27], we curated a list of 142 genes that encode proteins that participate in the MAPK signalling pathway which included genes involved in the ERK5 pathway (14 genes), the JNK pathway (52 genes), the p38 pathway (45 genes) and the ERK1/2 pathway (73 genes) (Supplementary File 1).

Next, we calculated the non-synonymous somatic mutation frequency (including single nucleotide mutations, short indels and insertions) for each of these genes across (1) all the samples and (2) each of the human cancer types represented among the 40,848 samples (Supplementary File 1). Here, samples of 15 out of the 198 cancer studies that we analysed were profiled by a targeted sequencing approach. These genes on the targeted sequencing panel of these studies included most of the well-known oncogenes (e.g., KRAS, BRAF, MAPK1), tumour suppressor genes (e.g., TP53 and TSC1), and the MAPK genes (e.g. NRAS, and NF1) that have been previously found frequently mutated in human cancer. Therefore, our calculated mutation frequencies of each gene involved the use of only the samples that were profiled for that specific gene.

Furthermore, we calculated the frequency of non-synonymous somatic mutations for groups of genes that participate in each of the four modules of the MAPK pathway (the ERK1/2 pathway, the ERK5 pathway, the JNK pathway, and the p38 pathway): firstly, across each of the cancer types and all the patient's samples (Supplementary File 1). Here, we allocated the samples that were profiled using targeted sequencing to the "undefined" groups if no mutations were observed in the targeted sequencing MAPK pathway's gene panel. This is because we could

not concretely ascertain that these samples have no gene mutations in any of the four MAPK pathway modules. Also, we calculated the frequency of somatic mutations for groups of genes that encode the various classes of MAPK proteins (e.g., MAPKKKs, MAPKs, DUSPs, and GTPases; see Fig. 1 and Supplementary File 1 for details), firstly across each of the cancer types and then across all of the patient's samples.

Finding the association between the patients' survival outcomes and the gene mutations of the MAPK pathway in the patients' tumours.

The Kaplan-Meier method[32] was used to compare the durations of overall survival (OS) and the durations of DFS between groups of patients that have tumours with versus without mutations in the MAPK pathway genes. Here, we also compared the OS and DFS durations for groups of patients with tumours that had: (1) mutations in genes of only one signalling module of the MAPK pathway; (2) mutations in genes of multiple MAPK signalling modules; and (3) with no mutations in genes of the MAPK pathways (see Fig. 1a and Supplementary File 2). In addition, we compared the OS and DFS durations for groups of patients with tumours that had mutations in genes that encode: (1), only one class of MAPK pathway proteins (e.g., MAPKKKs, or MAPKs); (2) multiple classes of the MAPK pathway proteins; and (3) with no mutations in genes that encode any MAPK pathway proteins (see Fig. 1b and Supplementary File 2). Note: we conducted all the survival analyses without considering any of the covariates that are likely to influence the OS and DFS outcomes of the cancer patients. Also, we exclude from our survival analyses, the sample of the "undefined" groups (unknown mutation status of the MAPK pathway module or MAPK pathway genes that encode specific MAPK proteins).

**Dependence of cell lines on MAPK signalling pathway genes**. We obtained data from the Achilles project at the DepMap Portal version 19Q4[20] on the fitness of 688 cell lines derived from 35 different human cancer types following CRISPR knockouts of 18,333 individual genes. See https://depmap.org/portal/ for information on the Achilles CRISPR-derived gene dependency descriptions.

In brief, regarding the CRISPR-derived gene effects: "a lower score means that a gene is more likely to be dependent in a given cell line. A score of 0 is equivalent to a gene that is not essential, whereas a score of −1 corresponds to the median of all "common essential" genes".

Within the database, the genes are grouped into four primary categories based on observed cell line fitness after CRISPR-mediated gene knockouts as follows:

- Common essential genes are those genes "which, in a large, pan-cancer screen, rank in the top X most depleting genes in at least 90% of cell lines. X is chosen empirically using the minimum of the distribution of gene ranks in their 90th percentile least depleting lines".
- Strongly selective genes are those "whose dependency is at least 100 times more likely to have been sampled from a skewed distribution than a normal distribution (i.e. skewed-LRT value >100)".
- Essential genes are those which are associated with cell fitness in only one or a few cell lines, but whose dependency is <100 times more likely to have been sampled from a skewed distribution than a normal distribution (i.e., less than that of the strongly selective genes).
- Non-essential genes are those which show no effect on cell fitness in any of the 688 tested cell lines.

Here, we sought to evaluate the extents to which different cell lines are dependent on MAPK signalling genes for their fitness. First, to unearth the cell line dependencies on genes from each MAPK signalling module, we calculated the percentage of genes within each MAPK pathway module that are "strongly selective" or "common essential" across the cancer types that are represented by the cell lines, and across all the cell lines (Supplementary File 3). Furthermore, to reveal the dependencies of cell lines on specific classes of MAPK pathway genes (e.g., GTPases, MAKKKs, and MAPKs), we calculated the percentage of genes that encode various classes of MAPK pathway proteins. Here, we only used genes that categorised as either "strongly selective" or "common essential" across the cell lines and cancer types that are represented by these cell lines in the Achilles database.

**Comparison of dependency of cell lines on oncogenes and tumour suppressor genes**. We processed the Achilles CRISPR-derived gene dependency data by first annotating the oncogenes and TSGs using information from multiple sources. These included: (1) the Sanger Consensus Cancer Gene Database[74] (699 oncogenes and TSGs); (2) the UniProt Knowledgebase[75] (304 oncogenes and 741 TSGs); (3) the TSGene database[76] (1220 TSGs); and (4) the ONGene database[77] (725 oncogenes). We collated datasets from these four sources to yield a list of 3688 known oncogenes and TSGs, representing 2932 unique genes (1021 Oncogenes and 1911 TSGs). We then used the list of oncogenes and TSGs to extract a list of: (1) MAPK pathway genes that are oncogenes; (2) MAPK pathway genes that are TSGs; (3) oncogenes that are not MAPK pathway genes; and (4) TSGs that are not MAPK pathway genes. Next, we compared the mean CRISPR-derived gene dependence scores for these four groups of genes using a one-way analysis of variance (Supplementary Fig. 3b).

**The essential MAPK pathway genes across cell lines**. We counted the number of instances in which the CRISPR-derived dependence score of each gene within each cancer cell line was <−0.5 (the cut-off point that was devised by the Achilles project to denote reduced cell fitness after CRISPR-mediated gene knockouts) to find the number MAPK pathway genes that are associated with a reduction in cell fitness across all the cell lines.

**The dependence of cancer cells on each of the MAPK pathway genes**. We sought to identify precisely which human cancer types (as represented by the cell lines) are significantly more dependent on oncogenes of each MAPK pathway module compared to all other oncogenes. Here, we grouped the cell lines into categories based on their primary tissue of origin. Then, focusing only on the oncogenes of each MAPK signalling module, for each cancer type we compared the mean CRISPR-derived dependence score of the oncogenes that were members of each of the MAPK signalling modules to the pooled mean dependence scores of the MAPK pathway genes across all the cancer cell lines (Fig. 3c; Supplementary File 3). Furthermore, for each cancer type, we compared the mean CRISPR-derived dependence score of the oncogenes for each MAPK signalling module to the mean dependence score of the non-MAPK pathway oncogenes (Supplementary Fig. 4a).

Additionally, we compared mean CRISPR-derived dependence scores between genes that encode classes of MAPK pathway proteins for each cancer type to the mean pooled dependence scores of all other MAPK pathway oncogenes for all other cancers (Supplementary Fig. 4b). Finally, for each different cancer types we compared the mean dependence scores of genes that encode different classes of MAPK pathway proteins to the pooled mean dependence scores of all non-MAPK pathway genes (Supplementary Fig. 4c).

**Hierarchical clustering of CRISPR fitness data of the cell line**. To compare patterns of gene dependencies between the 688 cell lines, we applied unsupervised hierarchical clustering with a cosine distance metric using complete linkage to CRISPR-derived dependence scores of the MAPK pathway genes.

Relationship between Achilles CRISPR-based fitness screens and the transcription profiles, methylation profiles and copy number variation profiles of cell lines.

We used the clustergram depicting the similarities and differences between the CRISPR-derived MAPK pathway gene dependence scores of the different cell lines to visualise the relationships between these dependence scores and: (1) the mRNA transcription profiles of the cell lines; (2) the DNA methylation profiles of the cell lines; and (3) the copy number variation profiles of the cell lines. Since the CCLE has performed comprehensive molecular profiling of the cell lines that are represented within the Achilles datasets, we retrieved the mRNA transcription, DNA methylation and copy number variation data from this database. We then arranged the genes represented in these datasets so that their order corresponds with the pattern of clustering that was produced using the CRISPR-derived dependence scores for each gene (Fig. 4a).

**Correlation between genes essentialities and transcriptional signatures**. We applied unsupervised hierarchical clustering with a cosine distance metric using complete linkage to CRISPR-derived dependence scores of all 18,023 genes (Supplementary Fig. 5a) to reveal the clustering of these genes across cell lines. Next, for these 18,023 genes, we retrieved data on: (1) the mRNA transcription levels of the genes in 667 cancer cell lines from the CCLE; (2) the mRNA transcription levels of these genes in 10,967 primary cancer samples from the TCGA database[18]; (3) and the transcription levels of these genes in 53 normal human tissues measured in over 15,000 healthy individuals from GTEx consortium[78]. We then arranged the genes in columns according to the clustering pattern of these genes based on their CRISPR-derived dependency scores to visualize the relationships between the mRNA gene expression levels of normal tissues, primary tumours and cancer cell lines (see Fig. 5a; Supplementary Fig. 5).

Testing for an association between the MAPK pathway gene dependencies of cell lines and the responses of cell lines to MAPK pathway inhibitors.

From the GDSC database, we retrieved the dose-responses for 344 cancer cell lines of 30 different human cancer types to 28 drugs that target components of the MAPK pathway (Supplementary File 4)[21]. These 28 drugs are hereafter referred to as MAPK pathway inhibitors.

For each MAPK pathway gene (e.g., KRAS) we grouped the cell lines into two groups: those with a high CRISPR determined dependence on that gene (e.g., a KRAS dependence score < −0.5) and those with low dependence on that gene (e.g., KRAS dependence score >0.5). We then compared the IC50 values for each of the 28 MAPK pathway inhibitors between the two groups of cancer cell lines. Furthermore, for each of the MAPK pathway genes, we grouped the cell lines into another two groups: those with mutations in a particular gene (e.g., KRAS mutants) and those without mutations in that gene (e.g., cell lines with no KRAS mutations). Then we compared the IC50 values for each of the 28 MAPK pathway inhibitors between the two groups (i.e., mutant and non-mutant) of cell lines.

Next, we counted the number of MAPK pathway genes that were either "common essential" or "strongly selective" across each cell line. This gave us the absolute number of MAPK pathway genes to which each cell line is most dependent (Supplementary File 4). Here, we hypothesised that the cancer cell lines

whose fitness is highly dependent on the MAPK pathway genes would correspondingly exhibit a more robust response to the MAPK pathway inhibitors. We, therefore, split the cell lines that are represented in the GDSC database into two categories: (1) those which had more than the median number of MAPK pathway genes that are "common essential" or "strongly selective" (these are the cell lines with a higher MAPK pathway gene dependence) and (2) those which had fewer than the median number of MAPK pathway genes that are "common essential" or "strongly selective" (these are the cell lines with a lower MAPK pathway gene dependence). Next, we compared mean IC50 values between these two groups of cell lines of the 28 MAPK pathway inhibitors (Supplementary File 4).

We grouped the cell lines that are represented in the GDSC database based on their primary tissue of origin. For each of these groups, we then calculated the median number of MAPK pathway genes in the "common essential" or "strongly selective". We used this median value as a cut-off point to classify the cohort of cancer types represented by the cell lines into two categories: those cancer types with higher than the median number of "common essential" or "strongly selective" MAPK pathway genes, and those with fewer than the median number of such genes. We then compared the mean dose-responses between the cell lines in these two groups to each of the 28 MAPK pathway inhibitors (Supplementary File 40).

**Enrichment analysis**. We performed gene set enrichment analysis for specific Gene Ontology Biological Processes terms by querying Enrichr with the genes that showed a Pearson's correlation coefficient between self-mRNA and the CRISPR-derived dependence score of either > 0.3 or <−0.3. (see Supplement File 3 and Supplementary Fig. 5b, c)[79].

**Association between MAPK pathway gene dependencies and mRNA transcription profiles following MAPK pathway inhibitor treatment**. From the Gene Expression Omnibus (GEO; https://www.ncbi.nlm.nih.gov/geo/query/acc.cgi?acc=GSE101406), we obtained mRNA transcription responses profiled by the LINCS project[80] for six cancer cell lines after small molecule inhibitor perturbations. The elements of these data include the names and concentrations of the anti-cancer drugs used to treat the six cell lines and the mRNA transcription responses following drug treatment. Next, we used the Connective Map toolbox[81] in MATLAB to retrieve only the cancer cell lines with corresponding dose-response profiles in the GDSC and CCLE database to the seven MAPK pathway inhibitors that are represented within the LINCS dataset that we retrieved. We found that only the cell lines, MCF7 and A549 when treated with the MEK inhibitors, selumetinib and PD-0325901 were common between the databases. We, therefore, evaluated the MAPK pathway mRNA transcription signatures that occur after selumetinib and PD-0325901 treatment in these two cell lines cognisant of the fact these cell lines have different dose-response profiles to selumetinib and PD-0325901 (see Fig. 7 and Supplementary Fig. 7).

**Statistics and reproducibility**. We performed all statistical analyses in MATLAB 2019b. Where appropriate, we used the independent sample Student $t$-test, Welch test, the Wilcoxon rank-sum test and the one-way Analysis of Variance to compare groups of continuous variables. All statistical tests were considered significant if the returned two-sided $p$-value was <0.05 for single comparisons. Correcting for the multiple hypotheses test was done by calculating a two-sided $q$-value (false discovery rate) for each group/comparison using the Benjamini & Hochberg procedure.

**Ethics approval**. The study protocol was approved by The University of Cape Town; Health Sciences Research Ethics Committee IRB00001938. The publicly available datasets were collected by the cBioPortal, TCGA, CCLE, Achilles, GDSC, and LINCS projects and made available via their respective project databases. The methods used here were performed following the relevant policies, regulations and guidelines provided by the TCGA, CCLE, DepMap, GDSC, and LINCS projects.

**Reporting summary**. Further information on research design is available in the Nature Research Reporting Summary linked to this article.

## Data availability

The data that support our results are available from the following repositories: cBioPortal; https://www.cbioportal.org/, the Genomics of Drug Sensitivity in Cancer; https://www.cancerrxgene.org/, the Cancer Cell Line Encyclopaedia; https://portals.broadinstitute.org/ccle/data, the LINCS project; http://www.lincsproject.org, the Genotype-Tissue Expression project; https://gtexportal.org/home/, the COSMIC Consensus Cancer Genes; https://cancer.sanger.ac.uk/census, and the Project Achilles; https://depmap.org/portal/. The pre-processed dataset can be found in the Supplementary Data and are named as follows: Supplementary Data 1: Supplementary data of cancer studies and mutations of the MAPK pathway genes; Supplementary Data 2: Clinical outcomes across various groups; Supplementary Data 3: Achilles fitness screens across the cancer cell lines of various cancer types; Supplementary Data 4: Dose-response profiles of the cancer cell lines as profiled the GDSC and associated results of the various statical test.

## Code availability

Custom code written in MATLAB for processing and analysis of the data presented here is freely available at https://github.com/smsinks/Integrated-Analysis-of-MAPK-Pathway-Across-Human-Cancers and at http://doi.org/10.5281/zenodo.4274507[82]. The repository includes some pre-downloaded datasets and conversion files required for the analysis.

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

## Acknowledgements

Student bursary funding for this project was provided by H3ABioNet, supported by the National Institutes of Health Common Fund under grant number U24HG006941. The content of this publication is solely the responsibility of the authors and does not necessarily represent the official views of the National Institutes of Health.

## Author contributions

The study was conceptualized by M.S., P.N., and D.P.M. The methodology was designed by M.S., N.M., P.N., and D.P.M. M.S. and P.N. performed the formal analysis of the data. M.S., P.N., and D.P.M. drafted the manuscript. Editing and reviewing of the manuscript was carried out by M.S., N.M., P.N. and D.P.M. Data visualisations were produced by M.S. M.S. was supervised by N.M. and D.P.M.

**Competing interests**

The authors declare that they have no competing interests.

**Additional information**

