## [Peer Review File · Communications Biology]

Reviewers' comments:

Reviewer #1 (Remarks to the Author):

The authors have done a comprehensive study of the MAPK pathway in multiple cancers, but I am unsure what is novel about the paper. It is a large study, so maybe the value is just confirming what we know in such a large dataset.

1. How is this new data, we know that KRAS and BRAF are the most common mutations of this pathway?
2. The section on outcomes is hard to follow, why only show OS results and not DFS as well for all analysis?
3. The section on fitness of cells is too vague and needs to be more specific about the genes affected.
4. How is this novel, showing that inhibitors work better in cell lines depended on the pathway? We already know that.
5. Discussion – did we not already know they were the most frequently mutated genes in cancer?
6. Discussion paragraph 2 should say worse overall survival outcomes as you saw no difference in DFS.
7. Figures appear too complex at present to follow at present, and need simplified or better headings.

Reviewer #2 (Remarks to the Author):

Integrated Molecular Characterization of the MAPK Pathways in Human Cancers

The manuscript submitted by Dr Musalula Sinkala and his group is interesting. It is an arduous work of analysis of data banks of the expression of MAPK pathways in different types of cancer and its possible involvement in chemo resistance. The manuscript is well written in general and easy to follow. It is required to attend some observations indicated below.

Major observations:

-Check the spelling in the title: Characterization.

- Introduction: Requires integration, short paragraphs of topics related to the field are included, but they are not integrated with each other and do not help to highlight the importance of the study.

-Results:

- page 6 refers supplementary file 1 regarding what is mentioned "The percentage 176 of tumors with mutations ranged from 0% in small cell carcinomas of the ovary to 177 100% in rectal and esophagogastric cancers". However, it is difficult to observe this result in the Excel file. We recommend a graph that could be inserted in the same Excel file to show the results (Pag 6 line 175).

-Order description of figure 5 in the paragraph does not correspond to figure 5. Lines 355-364, page 11-12).

-The figure that concludes what is mentioned on line 389-391 page 12-13 is not cited or shown.

"Again, we then compared the dose-response of the two groups using drug response data from the GDSC. Altogether these two sets of comparisons revealed two critical insights that are of relevance to the use of MAPK pathway inhibitors as cancer therapeutics"

-Check that the number of figures corresponds; line 555, page 18 "pathway modules (Figure 2d and Figure 3e)"

Minor observations:

-The figures do not have numbers and it was not easy to follow them. The text refers to the letters of

the figures with capital letters and the figures are with lower case letters.

-Supplementary figure 6 legend, mentions (d), but supplementary figure 6d is not shown, (refers to (c)?)

Reviewer #3 (Remarks to the Author):

Sinkala et. al. present a very comprehensive analysis of the MAPK pathway across multiple cancer datasets, both from tumor samples as well as cell lines. The paper has some interesting results but some parts of the analyses are either unclear or need to account for other variables.

Major comments

1 - It is unclear from the methods section whether the authors have looked only at somatic SNVs or if they have looked at CNVs, truncating mutations etc. Also, it is not clear which missense mutations they have included: is it all mutations in MAPK genes, only those that are confirmed to be oncogenic (i.e. from OncoKB or Bailey et. al. Cell 2018, for example) etc? The authors should make that more clear. Also, is the information for both CNVs and missense mutations available for all samples? I am asking because not all samples from cBioPortal have been analyzed with the same platforms, making the coverage of different genes uneven.

2 - It is also not clear whether authors have included in their analyses the cancer type and the age of the patients in their overall survival and disease-free survival analyses. Given the widely different survival times, treatment options and MAPK pathway mutation frequencies across cancer types, including this as a covariate in the analysis is critical.

3 - I think a supplementary figure with an UpSet plot showing which pathways are mutated in the same samples in the "Multiple pathways" group of Figure 2a would be very useful to have a better glimpse of the mutation frequencies, mutual exclusivity and co-occurrence of mutations across the different MAPK subpathways.

For example, there seems to be a near-perfect correlation between p38 and JNK mutation frequencies (figure below generated with the data from supplementary table 1). Given the wide differences in survival outcomes of these two pathways (Figure 2f), this correlation seems either very unlikely or driven by the few tumors that are not mutated in both pathways. In either case this is something interesting that is not clear from the current paper Figures.

4 - The global negative correlation between dependency scores and mRNA expression (Figure 4b) might be statistically significant, but the distribution is far from normal. In fact, it has a very strange shape, suggesting that it is driven by other properties. I think it might make more sense to show the overall distribution of the correlation coefficients of the different genes, highlighting the ones that are statistically significant (you might even consider grouping them by MAPK subpathway) as the authors do in Figure 5. For instance, Figures 4c and 4e are much more informative, and the authors themselves focus on the subset of "common essential genes" for the rest of Figure 4 rather than on the global correlation.

5 - In the analyses showing the correlation between mRNA and dependency scores, the authors show

various examples where the driving factor seems to be either the cancer type or the mutation status of the gene, rather than mRNA expression itself, such as in KRAS. Have the authors included these two important covariates in their analysis? They have looked in detail at these two features for the drug response analysis (Figure 6), but I don't see any reference to a multivariate analysis in Figure 4.

Minor comments

1 - Including TP53 as a member of the MAPK pathway, while technically correct, I think might be biasing some of the results, particularly those that look at multi-pathway mutated samples regardless of the gene. For example, TP53 is a marker of worse outcomes in many cancer types. Do the survival analyses still hold true if one looks at samples with mutations in MAPK pathway in other genes than TP53? Something similar happens with the first paragraph of the discussion where the authors claim that MAPK pathway is the second most mutated in cancer. Is this still true without TP53? I'd suggest to take a look at those results to ensure that they are not driven by TP53 alone.

Another example: the survival analysis of the patients according to the position of the pathways that are mutated reveals that patients with mutation in the "substrate" proteins have the worst outcomes (Figure 2h). There is an almost perfect correlation between the frequency of TP53 mutations and "substrate" mutations, suggesting that most of the signal of the survival analysis is coming only from TP53 (figure below, generated with data from supplementary table 1).

2 - If TP53 is a substrate and it's mutated in 34% of all patients in the study (page 5, second paragraph), how is it possible that substrates, overall, are only mutated in 32% of all patients (Figure 1, left)? In the same line, if TP53 is mutated in 34% of all patients, how is it that p38 pathway is also mutated only in 34% of all patients? If a sample has a mutation in a gene of this pathway, is there always a second mutation in TP53 in the same sample? This seems to be the case for most cancer types (see the Figure below which I made with your data). If that is the case I think calling the signal to come from the "p38 pathway" is a bit misleading, since most of what you are capturing are TP53 mutations, which are a world on their own.

3 - I think that Figure 5 points to an interesting story, but I don't see the connection with MAPK pathways. Are the oncogenes and transcription factors that the authors highlight part of MAPK? Do genes in the MAPK pathway have a stronger correlation between mRNA and dependency scores than the rest of the genes?

4 - I think many Figures would benefit from clustering. For example in Figure 6a, 7b and 7c genes and cells / treatments are sorted alphabetically, making it very difficult to see any patterns in the data.

Review Response: Integrated Molecular Characterisation of the MAPK Pathways in Human Cancers

We want to thank the reviewers for evaluating our study and providing their insightful and constructive suggestions regarding several aspects of the manuscript. As described below, we have addressed the concerns raised by all the reviewers and made modifications to the manuscript, as suggested.

Please find in this letter a "point-by-point" response to each comment. We have taken all the Reviewer's comments seriously and have revised the manuscript accordingly (redline version). The comments (*italics*) and the changes made in response (blue) and the current Page and Line numbers highlighted in yellow are listed below. References (cyan) made to any new figures, edited figures and supplementary files.

Reviewer #1 (Remarks to the Author):

The authors have done a comprehensive study of the MAPK pathway in multiple cancers, but I am unsure what is novel about the paper. It is a large study, so maybe the value is just confirming what we know in such a large dataset.

We thank the Reviewer for appreciating our study and providing supportive critiques. We have addressed the suggestions in a pointwise manner as enlisted below.

Comment 1: *How is this new data, we know that KRAS and BRAF are the most common mutations of this pathway?*

Response: Yes, we know that KRAS and BRAF are the most common mutations of this pathway. However, no large-scale studies have ever been conducted to evaluate KRAS and BRAF mutations across so many cancer types across a comparable number of studies. Also, we have shown that among the 101 cancer types that we analysed, the extent of mutations in KRAS, BRAF - and among many other MAPK pathway genes - varied widely between different cancer types. For example, we found no BRAF mutations in 35 cancer types, including Acute Lymphoblastic Leukemia (210 samples), Kidney Chromophobe (65 samples), Acute Myeloid Leukemia (165 samples) and Adenoid Cystic Carcinoma (212 samples). Furthermore, we found no KRAS mutations in 33 cancer types of various tissues of origin, including Adenoid Cystic Carcinoma (212 samples), Esophageal Squamous Cell Carcinoma (231 samples), Hepatocellular Carcinomas (383 samples), Small Cell Lung Cancer (216 samples) and Neuroblastoma (383 samples). These mutations also vary across tumours of the same organ/tissue, for example, we found no KRAS mutations in Pancreatic Neuroendocrine Tumors (118 samples), but found KRAS mutations in 85% of all Pancreatic Adenocarcinomas (759 samples). While not wishing to diminish the perceived importance of KRAS and BRAF, we have attempted to indicate the true scale and complexity of the cancer associated MAPK pathway gene mutational landscape: which is something that we think is novel.

Comment 2: *The section on outcomes is hard to follow, why only show OS results and not DFS as well for all analysis?*

Response: We have now included DFS analyses for all the corresponding OS results. Furthermore, we have edited the section on outcomes to make it easier to follow.

Comment 3: *The section on the fitness of cells is too vague and needs to be more specific about the genes affected.*

Response: We have now edited the section on cell fitness to make the results section clearer and more straightforward to follow. We have also included Supplementary File 3, which details the specific MAPK genes that are affected, including the number of cell lines containing affected genes, together with the correlation between the mRNA transcription abundance of each gene and the CRISPR-derived Achilles cell fitness scores associated with those genes.

Comment 4: *How is this novel, showing that inhibitors work better in cell lines depended on the pathway? We already know that.*

Response: Our analyses are novel specifically because we show that this is not so obvious. Specifically, we initially expected that inhibitors would work better in cell lines that are dependant either on the targeted pathway as a whole, or on particular targeted genes of the pathway. However, we paradoxically found that in multiple instances where cell lines are dependent on MAPK genes, these cell lines are resistant to small molecule inhibitors of MAPK pathway proteins. For example, we find that cell lines that are highly dependent on, among others, PTPN11, DUSP13, DUSP8, and TRAF6, are all significantly more resistant to various MAPK pathway inhibitors (please see Figure 6a). Furthermore, we also found that the CRISPR-derived dependence scores of 18 other genes (including RAF1 and MAP3K2) were associated with mixed responses; i.e., significantly increased sensitivity to some of the inhibitors and significantly decreased sensitivity to others (please refer to Supplementary File 1, Sheet: "Cancer-MAPK Gene Mutations").

Comment 5: *Discussion – did we not already know they were the most frequently mutated genes in cancer?*

Response: We knew that they were the most mutated in several cancer types but did not have comprehensive information on the MAPK pathway mutations across such a large number (101) of cancer types.

Comment 6: *Discussion paragraph 2 should say worse overall survival outcomes as you saw no difference in DFS.*

Response: Thank you for noticing this error. We have now corrected this statement to "worse overall survival".

Comment 7: *Figures appear too complex at present to follow at present and need simplified or better headings.*

Response: While we would want to provide simplified figures, the other Reviewers appear to indicate that they would wish to see figures that show added multivariate analyses. We have thus given better headings to most of the figures in an attempt to better clarify the existing figures.

Reviewer #2 (Remarks to the Author):

The manuscript submitted by Dr Musalula Sinkala and his group is interesting. It is an arduous work of analysis of data banks of the expression of MAPK pathways in

different types of cancer and its possible involvement in chemoresistance. The manuscript is well written in general, and easy to follow. It is required to attend some observations indicated below.

Response: We thank the Reviewer for appreciating various aspects of this manuscript. We also welcome the constructive comments and suggestions that have been offered to improve the manuscript further. We have addressed the points raised by the Reviewer below.

Major observations:

Comment 1: Check the spelling in the title: Characterization.

Response: We thank the Reviewer for this correction. However, we have retained the British spelling ("Characterisation") since the remainder of our manuscript is written in British English.

Comment 2: *Introduction: Requires integration, short paragraphs of topics related to the field are included, but they are not integrated with each other and do not help to highlight the importance of the study.*

Response: We have now integrated the section of the introduction and a new paragraph that hopefully now better highlights the importance of our study.

Comment 3: *Page 6 refers supplementary file 1 regarding what is mentioned "The percentage of tumors with mutations ranged from 0% in small cell carcinomas of the ovary to 100% in rectal and esophagogastric cancers". However, it is difficult to observe this result in the Excel file. We recommend a graph that could be inserted in the same Excel file to show the results (Pag 6 line 175).*

Response: We have now inserted a graph in the Supplement File 1 - Sheet "MAPK Alteration Across Cancers" showing the frequency of MAPK pathway gene mutations within each cancer type.

Comment 4: *Order description of figure 5 in the paragraph does not correspond to figure 5. Lines 355-364, page 11-12).*

Response: We have now corrected the order of description of Figure 5.

Comment 5: *The figure that concludes what is mentioned on line 389-391 page 12-13 is not cited or shown. "Again, we then compared the dose-response of the two groups using drug response data from the GDSC. Altogether these two sets of comparisons revealed two critical insights that are of relevance to the use of MAPK pathway inhibitors as cancer therapeutics".*

Response: We have now cited (made reference to the figure) and shown the figure of the results in the section as mentioned above of our paper (see Figure 6a).

Comment 6: *Check that the number of figures corresponds; line 555, page 18 "pathway modules (Figure 2d and Figure 3e)".*

Response: We have now referenced the corrected figure panels in the mentioned sentence.

Minor observations:

Comment 7: *The figures do not have numbers, and it was not easy to follow them. The text refers to the letters of the figures with capital letters, and the figures are with lower case letters.*

Response: We have now given better descriptive names to all the figures and corrected the references to the figures within the text. Now all figures are referred to with lower case letters and are referenced by lower case letters in the text.

Comment 8: *Supplementary figure 6 legend, mentions (d), but supplementary figure 6d is not shown, (refers to (c)?)*

Response: Thank you for the observation. We have now corrected this.

Review #3: Remarks to Authors.

Sinkala et al. present a very comprehensive analysis of the MAPK pathway across multiple cancer datasets, both from tumor samples as well as cell lines. The paper has some interesting results but some parts of the analyses are either unclear or need to account for other variables.

Response: We thank the Reviewer for appreciating various aspects of this manuscript. We also welcome the constructive comments and suggestions that have been offered by the Reviewer to improve the manuscript further. We have addressed the points raised by the Reviewer as below.

Major comments

Comment 1: *It is unclear from the methods section whether the authors have looked only at somatic SNVs or if they have looked at CNVs, truncating mutations etc. Also, it is not clear which missense mutations they have included: is it all mutations in MAPK genes, only those that are confirmed to be oncogenic (i.e. from OncoKB or Bailey et. al. Cell 2018, for example) etc? The authors should make that more clear. Also, is the information for both CNVs and missense mutations available for all samples? I am asking because not all samples from cBioPortal have been analyzed with the same platforms, making the coverage of different genes uneven.*

Response: We have now categorically stated in the methods section that we used only non-synonymous somatic mutation frequencies, including single nucleotide variations, short indels and insertions. Therefore, we excluded synonymous mutations from our analyses. We have not looked at only mutations that are confirmed to be oncogenic because the mechanisms of resistance (increased cell fitness after drug and/or genetic perturbation) are influenced by both non-oncogenic and oncogenic changes¹⁻³. We did not use both CNVs and missense mutations for two reasons: 1) CNVs are not available for all samples and 2) we intend to explore the relationship between copy number changes, or DNA methylation profiles, micro RNAs, Achille cell fitness screens and drug sensitivities in a future study.

Comment 2: *It is also not clear whether authors have included in their analyses the cancer type and the age of the patients in their overall survival and disease-free survival analyses. Given the widely different survival times, treatment options and MAPK pathway mutation frequencies across cancer types, including this as a covariate in the analysis is critical.*

Response: We had initially intended to perform survival analyses corrected for the various patient and sample covariates that are available in the clinical data. However, we found that selecting the most informative clinical characteristics for inclusion in our survival model, and comparing these characteristics, is by itself

convoluted. For examples, the clinical data of these samples include over 100 covariates that we could have in our survival model. These include, among others, age, sex, cancer type and cancer subtypes within each cancer types (both the histological and molecular subtypes), ethnicity, mutation counts, clinical disease state, cancer stage, types of treatment, lymph node involved, metastasis, site of metastasis, race, radiation therapy, other gene mutations that co-occur with those in specific MAPK pathway modules (even those within pathway modules, e.g., the TP53 mutations that you thankfully highlighted), blood cell parameters (e.g., lymphocyte counts), and the underlying medical conditions. Because of these complications, we settled on applying the Kaplan-Meier method for our survival analysis for 1) its advantage that the median survival times are unadorned, and the test statistics are unambiguous and straightforward to interpret⁴⁻⁷ and 2) selected this method upon studying the literature— most pan-cancer studies (including those of The Cancer Genome Atlas) have used, and/or have encouraged the use of, the simple Kaplan-Meier analysis and log-rank test^{6,8-14}. Furthermore, while we take the Reviewer's suggestion very seriously, to conduct our survival analyses while correcting for various covariates would constitute a vast statistical task, for which the interpretation and reporting of findings would necessitate a separate publication such as those by Liu et al.⁶ and Huo et al.⁵. We are aware that an approach as applied here exposes a significant limitation of survival analysis in pan-cancer studies, but such analyses may also be the best compromise when our objective is to provide an overview of survival outcomes in the face of such high-dimensional datasets.

Comment 3: *I think a supplementary figure with an UpSet plot showing which pathways are mutated in the same samples in the "Multiple pathways" group of Figure 2a would be very useful to have a better glimpse of the mutation frequencies, mutual exclusivity and co-occurrence of mutations across the different MAPK subpathways. For example, there seems to be a near-perfect correlation between p38 and JNK mutation frequencies (figure below generated with the data from supplementary table 1). Given the wide differences in survival outcomes of these two pathways (Figure 2f), this correlation seems either very unlikely or driven by the few tumors that are not mutated in both pathways. In either case this is something interesting that is not clear from the current paper Figures.*

Response: We have now included an UpSet plot showing which pathway associated genes are mutated in the samples in the "multiple pathways group of Figure 2a (please see Supplementary Figure 2b, also shown below). Here, we have noted, as suggested by the Reviewer, the significantly strong correlation between TP53 mutations and the frequencies of JNK pathway gene mutations, TP53 mutations and p38 pathway gene mutations, which results in a significant correlation in the mutations frequencies between the JNK and p38 pathways. Therefore, we have excluded TP53 mutations in our calculations of all mutation frequencies in our survival analyses. Here, after excluding TP53 mutations from our analyses, we found that only 505 tumours have mutations in genes of both the p38 and JNK pathways. The overlap between samples is much lower – compared to the initial overlap of 8,107 tumours with mutations in both p38 and JNK pathway genes when we include TP53 mutations.

Interestingly, even after excluding TP53 mutations from our overall survival and disease-free survival analysis groupings, we still found that patients with tumours that exclusively have JNK pathway mutations (1,545 patients) exhibited better outcomes than those patients with tumours that only have p38 mutations (845

patients). Furthermore, we have now included a paragraph in the discussion section where we attempted to highlight why tumours with only JNK pathway mutations exhibited significantly better overall survival outcomes (line 593 to 598): "Given that the activation of JNK signalling has a tumour suppressor role¹⁶⁻¹⁷ in many contexts (but a tumour promotor role in others contexts¹⁶⁻¹⁸), the enhanced survival of patients with tumours that have JNK pathway gene mutations suggests that, whenever these mutations have any impact on cancers at all, they may most commonly promote anti-tumour activity". Also, within the results sections, we have now included the following statement to point the readers to some of the literature which shows the association between the JNK pathway and enhanced apoptosis leading to improved treatment outcomes (line 214 to 218): "Concerning the OS periods, patients with tumours that had mutations in JNK pathway genes (median survival = 141.7 months), had the most favourable outcomes, a finding that is consistent with those of other studies which show an association between the JNK pathway alterations and enhanced apoptosis²⁰⁻²² and improved survival outcomes^{20,21,23}".

Figure 1: Upset plot showing the mutual exclusivity and co-occurrence of mutations across the different MAPK pathway modules. Note that TP53 mutations are excluded from the plotted data

Comment 4: The global negative correlation between dependency scores and mRNA expression (Figure 4b) might be statistically significant, but the distribution is far from normal. In fact, it has a very strange shape, suggesting that it is driven by other properties. I think it might make more sense to show the overall distribution of the correlation coefficients of the different genes, highlighting the ones that are statistically significant (you might even consider grouping them by MAPK subpathway) as the authors do in Figure 5. For instance, Figures 4c and 4e are much more informative, and the authors themselves focus on the subset of "common essential genes" for the rest of Figure 4 rather than on the global correlation.

Response: We have replaced Figure 4b with a panel plot showing the overall distribution of Pearson's correlation coefficients that we found statistically significant. We have also included in Supplementary File 3 the actual Pearson's correlation coefficients for each gene and their associated p-values (see the Excel sheet named "MAPK- CRISPR vs mRNA Corr" of the Supplementary File 3).

Comment 5: In the analyses showing the correlation between mRNA and dependency scores, the authors show various examples where the driving factor

seems to be either the cancer type or the mutation status of the gene, rather than mRNA expression itself, such as in KRAS. Have the authors included these two important covariates in their analysis? They have looked in detail at these two features for the drug response analysis (Figure 6), but I don't see any reference to a multivariate analysis in Figure 4.

Response: We are unsure of what the Reviewer means here. We think that the Reviewer meant to say, "but I don't see any reference to a multivariate analysis in Figure [6]"? This is because Figure 4 showed the multivariate analyses that are referred to in comment 5. If such is the case, we have now replotted the data of Figure 6a to merge them with those of Figure S7a, i.e., now Figure 6a shows both the relationship between gene mutations and dependency scores to the drug response of the cell lines (See the new Figure 6a and 6b). We hope this sufficiently address the Reviewer's concern but remain open to a clarification here.

Minor comments

Comment 1 - Including TP53 as a member of the MAPK pathway, while technically correct, I think might be biasing some of the results, particularly those that look at multi-pathway mutated samples regardless of the gene. For example, TP53 is a marker of worse outcomes in many cancer types. Do the survival analyses still hold true if one looks at samples with mutations in MAPK pathway in other genes than TP53? Something similar happens with the first paragraph of the discussion where the authors claim that MAPK pathway is the second most mutated in cancer. Is this still true without TP53? I'd suggest to take a look at those results to ensure that they are not driven by TP53 alone.

Response: We have now excluded TP53 mutations from our survival analyses as stated in our response to Comment number 3. Interestingly, we found that survival analyses still hold -- we got similar findings albeit with different p-values and other test statistics. By excluding TP53 mutations from our analyses, we found that the number of samples with MAPK pathway gene mutations is 42%, a significantly lower number compared to our initial calculation of 58% samples if we include TP53 mutations. And yet still, the mutation frequency of 42% leaves the MAPK pathway genes as being the second most mutated in human cancers.

Comment 2: Another example: the survival analysis of the patients according to the position of the pathways that are mutated reveals that patients with mutation in the "substrate" proteins have the worst outcomes (Figure 2h). There is an almost perfect correlation between the frequency of TP53 mutations and "substrate" mutations, suggesting that most of the signal of the survival analysis is coming only from TP53 (figure below, generated with data from supplementary table 1).

Response: We have excluded TP53 from our survival analysis and found that indeed, as the Reviewer suggested would be the case, our estimated duration of the overall months are significantly increased from 55.4 months to 80.5 months for patients with tumours that only have mutations in the genes encoding substrate proteins (as shown in the new Figure 2h). Again, as indicated by the Reviewer, our previous estimate was in large part driven by TP53 mutations.

Comment 3: If TP53 is a substrate and it's mutated in 34% of all patients in the study (page 5, second paragraph), how is it possible that substrates, overall, are only

mutated in 32% of all patients (Figure 1, left)? In the same line, if TP53 is mutated in 34% of all patients, how is it that p38 pathway is also mutated only in 34% of all patients? If a sample has a mutation in a gene of this pathway, is there always a second mutation in TP53 in the same sample? This seems to be the case for most cancer types (see the Figure below which I made with your data). If that is the case I think calling the signal to come from the "p38 pathway" is a bit misleading, since most of what you are capturing are TP53 mutations, which are a world on their own.

Response: The confusion in these sentences and figure is essentially our own. The reported frequencies, captured only the mutations in tumour genes encoding substrate proteins (i.e., as in Figure 2a and Figure 2b). We have now correctly reported the frequency of these mutations in Figure 1 and edited the text in our paper accordingly. Furthermore, we have included the corrected frequencies (as shown in Figure 1) in Supplementary File 1 on the Excel sheets named "Cancer-MAPK pathways Mutations". On this sheet, we have also included the following description; "The frequency of mutations and the percentage of tumours that harboured mutations in the genes of each MAPK pathway module and each class of MAPK pathway genes (as shown in Figure 1). Here, the total number of samples and percentages shown do not add up to the total number of samples with mutations, they instead sum up to the total percentage of tumours with MAPK pathway gene mutations (as in Figure 2a and Figure 2b). This is because, for each sample, the mutations in each category were counted irrespective of other co-occurring mutations in the other categories, i.e., there is no "multiple pathways altered category" in this case (as in Figure 2a and Figure 2b)."

Comment 4: I think that Figure 5 points to an interesting story, but I don't see the connection with MAPK pathways. Are the oncogenes and transcription factors that the authors highlight part of MAPK? Do genes in the MAPK pathway have a stronger correlation between mRNA and dependency scores than the rest of the genes?

Response: In Figure 5, we wanted to assess our intriguing finding concerning the correlation between mRNA transcription and CRISPR-derived gene dependency by extending the analysis to include other genes. Also, we wanted to determine if, as in the cancer cell lines, this relationship still holds in healthy tissues and biologically perturbed tissues (including cancerous tissues).

Comment 5: I think many Figures would benefit from clustering. For example, in Figure 6a, 7b and 7c genes and cells / treatments are sorted alphabetically, making it very difficult to see any patterns in the data.

Response: We have now clustered the data that are plotted in Figure 6a, 7b and 7c.

References

1. Luo, J., Solimini, N. L. & Elledge, S. J. Principles of Cancer Therapy: Oncogene and Non-oncogene Addiction. *Cell* **136**, 823–837 (2009).
2. Blagosklonny, M. V. Cell Cycle Why Therapeutic Response May Not Prolong the Life of a Cancer Patient: Selection for Oncogenic Resistance Selection for Oncogenic Resistance. *Cell Cycle* **4**, 1693–1698 (2005).
3. Nagel, R., Semanova, E. A. & Berns, A. Drugging the addict: non-oncogene addiction as a target for cancer therapy. *EMBO Rep.* **17**, 1516–1531 (2016).
4. Hoadley, K. A. *et al.* Multiplatform analysis of 12 cancer types reveals molecular

- classification within and across tissues of origin. *Cell* **158**, 929–944 (2014).
5. Huo, D. *et al.* Comparison of Breast Cancer Molecular Features and Survival by African and European Ancestry in The Cancer Genome Atlas. *JAMA Oncol.* **3**, 1654–1662 (2017).
 6. Liu, J., Lichtenberg, T. & Hoadley, K. A. An Integrated TCGA Pan-Cancer Clinical Data Resource to Drive High-Quality Survival Outcome Analytics In Brief Analysis of clinicopathologic annotations for over 11,000 cancer patients in the TCGA program leads to the generation of TCGA Clinical Data Resource, which provides recommendations of clinical outcome endpoint usage for 33 cancer types. *Cell* **173**, 400-416.e11 (2018).
 7. Witten, D. M. & Tibshirani, R. Survival analysis with high-dimensional covariates. *Stat. Methods Med. Res.* **19**, 29–51 (2010).
 8. Wang, Y. *et al.* Comprehensive Molecular Characterization of the Hippo Signaling Pathway in Cancer. *Cell Rep.* **25**, 1304-1317.e5 (2018).
 9. Ge, Z. *et al.* Integrated Genomic Analysis of the Ubiquitin Pathway across Cancer Types. *Cell Rep.* **23**, 213-226.e3 (2018).
 10. Wang, Z. *et al.* lncRNA Epigenetic Landscape Analysis Identifies EPIC1 as an Oncogenic lncRNA that Interacts with MYC and Promotes Cell-Cycle Progression in Cancer Article lncRNA Epigenetic Landscape Analysis Identifies EPIC1 as an Oncogenic lncRNA that Interacts with MYC and Promotes Cell-Cycle Progression in Cancer. *Cancer Cell* **33**, 706-720.e9 (2018).
 11. Berger, A. C. *et al.* A Comprehensive Pan-Cancer Molecular Study of Gynecologic and Breast Cancers. *Cancer Cell* **33**, 690-705.e9 (2018).
 12. Chen, H., Li, C., Peng, X., Zhou, Z. & Weinstein, J. N. A Pan-Cancer Analysis of Enhancer Expression in Nearly 9000 Patient Samples Enhancer expression of ~9,000 tumors Tumors with global enhancer activation. (2018). doi:10.1016/j.cell.2018.03.027
 13. Schaub, F. X. *et al.* Pan-cancer Alterations of the MYC Oncogene and Its Proximal Network across the Cancer Genome Atlas. *Cell Syst.* **6**, 282-300.e2 (2018).
 14. Korkut, A. *et al.* A Pan-Cancer Analysis Reveals High-Frequency Genetic Alterations in Mediators of Signaling by the TGF- β Superfamily. *Cell Syst.* **7**, 422-437.e7 (2018).
 15. Dou, Y., Jiang, X., Xie, H., He, J. & Xiao, S. The Jun N-terminal kinases signaling pathway plays a 'seesaw' role in ovarian carcinoma: A molecular aspect. *Journal of Ovarian Research* **12**, (2019).
 16. Potapova, O., Basu, S., Mercola, D. & Holbrook, N. J. Protective Role for c-Jun in the Cellular Response to DNA Damage. *J. Biol. Chem.* **276**, 28546–28553 (2001).
 17. Johnson, G. L. & Nakamura, K. The c-jun kinase/stress-activated pathway: Regulation, function and role in human disease. *Biochimica et Biophysica Acta - Molecular Cell Research* **1773**, 1341–1348 (2007).
 18. Hess, P., Pihan, G., Sawyers, C. L., Flavell, R. A. & Davis, R. J. Survival signaling mediated by c-Jun NH2-terminal kinase in transformed B lymphoblasts. *Nat. Genet.* **32**, 201–205 (2002).
 19. Papachristou, D. J., Batistatou, A., Sykiotis, G. P., Varakis, I. & Papavassiliou, A. G. Activation of the JNK-AP-1 signal transduction pathway is associated with pathogenesis and progression of human osteosarcomas. *Bone* **32**, 364–371 (2003).
 20. Davila-Gonzalez, D. *et al.* Pharmacological inhibition of NOS activates ASK1/JNK pathway augmenting docetaxel-mediated apoptosis in triple-negative breast cancer.

- Clin. Cancer Res.* **24**, 1152–1162 (2018).
21. Fey, D. *et al.* Signaling pathway models as biomarkers: Patient-specific simulations of JNK activity predict the survival of neuroblastoma patients. *Sci. Signal.* **8**, ra130–ra130 (2015).
 22. Tarapore, R. S., Yang, Y. & Katz, J. P. Restoring KLF5 in esophageal squamous cell cancer cells activates the JNK pathway leading to apoptosis and reduced cell survival. *Neoplasia (United States)* **15**, 472–480 (2013).
 23. Bubici, C. & Papa, S. JNK signalling in cancer: In need of new, smarter therapeutic targets. *British Journal of Pharmacology* **171**, 24–37 (2014).

Reviewers' comments:

Reviewer #2 (Remarks to the Author):

Integrated Molecular Characterisation of the MAPK Pathways in Human Cancers

All comments were satisfactorily raised by the authors

Reviewer #3 (Remarks to the Author):

The authors did a very thorough job at addressing my comments. I have only two comments.

Regarding comment 1, which mutations are included in the analysis, the authors response about which mutations to include or not is perfectly reasonable. I still do not know, however, if the authors have checked whether all the samples from all the cancer types have been analyzed with WES. I am saying this because some patients in cBioPortal do not have WES data, but targeted sequencing. In this case, this could have an important impact in the mutation frequency of genes not included in the targeted sequencing panel as we could only detect mutations in a subset of all samples.

Regarding "Comment 2", including covariates in the survival analysis, while the answer provided by the authors is reasonable, I believe that it is necessary to add a comment or a sentence in the results section explaining that the survival analysis does not include covariates, so that readers are aware of this limitation.

Review Response: Integrated Molecular Characterisation of the MAPK Pathways in Human Cancers

First, we are grateful to the editor for the timely response and the reviewers for their useful comments and suggestions made to improve our manuscript. We tried our best to address their concerns and made modifications to the manuscript, as suggested. Also, we want to extend our appreciation for taking the time and effort necessary to provide such insightful guidance.

Please find in this letter a "point-by-point" response to each comment. We have taken all the Reviewer's Reviewer's comments seriously and have revised the manuscript accordingly (redline version). The comments (*italics*) and the changes made in response (blue) and the current Page and Line numbers highlighted in yellow are listed below. References (cyan) made to any new figures, edited figures and supplementary files.

Reviewers' Reviewers' comments:

Reviewer #2 (Remarks to the Author):

Comment 1: *All comments were satisfactorily raised by the authors*

Response: We are happy to have satisfactorily addressed all the comments and suggestions that were raised by the Reviewer.

Reviewer #3 (Remarks to the Author):

The authors did a very thorough job at addressing my comments. I have only two comments.

We thank the Reviewer for appreciating our study and providing supportive critiques. We have addressed the suggestions in a pointwise manner as enlisted below.

Comment 1: *Which mutations are included in the analysis, the authors response about which mutations to include or not is perfectly reasonable. I still do not know, however, if the authors have checked whether all the samples from all the cancer types have been analyzed with WES. I am saying this because some patients in cBioPortal do not have WES data, but targeted sequencing. In this case, this could have an important impact in the mutation frequency of genes not included in the targeted sequencing panel as we could only detect mutations in a subset of all samples.*

Response: We appreciate that the Reviewer has correctly noted that some of the samples were profiled using targeted sequencing. It turns out that this is a crucial point because the datasets downloaded from the cBioPortal using their Application Programming Interface returns data which misleadingly shows that all samples were profiled for all the MAPK pathways genes. Therefore, as suggested by the Reviewer,

this had a substantial impact on the mutation frequency of the genes not included in the targeted sequencing panel.

To address this problem, we have now manually extracted information of targeted sequencing panel for all the 15 cancer studies (out of the 192 cancer studies) which applied targeted sequencing to determine the mutation profile of the tumours samples. Please see the table below, which shows the abovementioned cancer studies.

Table 1: The Targeted Sequencing Cancer studies

cancerTypeId	name
{'blca_mskcc_solit_2014'}	{'Bladder Cancer (MSKCC, Eur Urol 2014)' }
{'brca_metabric' }	{'Breast Cancer (METABRIC, Nature 2012 & Nat Commun 2016)' }
{'coadread_mskcc' }	{'Colorectal Adenocarcinoma Triplets (MSKCC, Genome Biol 2014)' }
{'cscd_dfarber_2015' }	{'Cutaneous Squamous Cell Carcinoma (DFCI, Clin Cancer Res 2015)' }
{'gct_msk_2016' }	{'Germ Cell Tumors (MSKCC, J Clin Oncol 2016)' }
{'hcc_msk_venturaa_2018'}	{'Liver Hepatocellular Adenoma and Carcinomas (MSK, PLOS One 2018)' }
{'luad_tsp' }	{'Lung Adenocarcinoma (TSP, Nature 2008)' }
{'msk_impact_2017' }	{'MSK-IMPACT Clinical Sequencing Cohort (MSKCC, Nat Med 2017)' }
{'crc_msk_2017' }	{'Metastatic Colorectal Cancer (MSKCC, Cancer Cell 2018)' }
{'egc_msk_2017' }	{'Metastatic Esophagogastric Cancer (MSKCC, Cancer Discovery 2017)' }
{'lung_msk_2017' }	{'Non-Small Cell Cancer (MSKCC, Cancer Discov 2017)' }
{'prad_mskcc_2017' }	{'Prostate Cancer (MSKCC, JCO Precis Oncol 2017)' }
{'hnc_mskcc_2016' }	{'Recurrent and Metastatic Head & Neck Cancer (MSKCC, JAMA Oncol 2016)' }
{'scco_mskcc' }	{'Small Cell Carcinoma of the Ovary (MSKCC, Nat Genet 2014)' }
{'urcc_mskcc_2016' }	{'Unclassified Renal Cell Carcinoma (MSK, Nature 2016)' }

With the information concerning the targeted sequencing gene panel of each of these cancer studies. Using a custom script, we edited the mutation profiles of these cancer studies to omit genes that were not in the targeted sequencing panel. Interestingly, we found that the targeted sequencing panel of these studies included most of the well-known oncogenes (e.g. KRAS, BRAF, MAPK1), tumour suppressor genes (e.g. TP53 and TSC1), and the MAPK genes (e.g. NRAS, and NF1) that have been previously found frequently mutated in human cancer.

Accordingly, our calculated mutations frequencies of each MAPK pathway gene only included the samples that were profiled (sequencing) for that specific gene. Therefore, for each gene, the number of samples used to calculate the mutation frequencies of that specific gene vary by a small fraction.

However, this change does not lead to significantly different results from those which we previously presented because only a few samples were not profiled for some of the MAPK pathway genes. Also, in most case, the sequencing panels only excluded the infrequently mutated genes. For example, the mutation frequencies of many known cancer genes (e.g., TP53, KRAS and BRAF) remain unchanged. Please refer to the table below.

Table 2: Calculated Mutation Frequencies

Genes	Previous	Current
KRAS	9.99	9.99
BRAF	5.63	5.63
NF1	4.92	4.92
MAP3K1	2.71	2.73
NRAS	2.39	2.39
DUSP18	0.10	0.119
LAMTOR3	0.068	0.079
LAMTOR2	0.063	0.074
DUSP23	0.048	0.057
CDC42SE1	0.046	0.054

Furthermore, when calculating the frequencies of gene mutations within each MAPK pathway module, we have removed from the "not mutated group" the samples with no MAPK gene mutations which were not profiled for all the MAPK pathway genes (profiled using targeted sequencing) and created a new "undefined" group for those samples. Therefore, the number of samples with gene mutations in the MAPK pathways modules and the genes that encode MAPK pathway proteins classes are unchanged. Additionally, we have now included a new group of samples that we have named "undefined" – we don't concretely know that these samples do have MAPK pathway module (ERK1/2, p38, JNK or ERK5 pathways) mutations because some of the genes that were not sequenced could have mutations (See the pie chart). Please note that the "undefined" group of samples were previously included among the "Not Mutated" group. Therefore, all other statistics of the samples with mutations in the specific MAPK pathway modules and classes of MAPK proteins are unchanged.

Also, note that we have excluded the "undefined" in our survival analyses. Here, the omnibus p-values of group comparisons are now slightly different from those that we had previously presented. However, the multiple comparison p-values of the groups that do not involve a comparison with the "Not Mutated" group also remain unchanged.

We have also added new sections in the Methods section (Lines 663 – 670; 682 – 689; 689 – 695; 717 – 722) of our manuscript to clearly explain how we calculated the mutation frequencies as outlined here. We have also edited the panels of figure 2 and supplementary figure 2.

Comment 2: *About including covariates in the survival analysis, while the answer provided by the authors is reasonable, I believe that it is necessary to add a comment or a sentence in the results section explaining that the survival analysis does not include covariates so that readers are aware of this limitation.*

Response: We have now included a section in the results section and methods section explaining that the survival analysis does not include covariates that are likely to influence the overall survival and disease-free survival outcomes.

REVIEWERS' COMMENTS:

Reviewer #3 (Remarks to the Author):

The authors addressed all of my concerns